# NEXT VISUAL GRANULARITY GENERATION

**Yikai Wang[1], Zhouxia Wang[1], Zhonghua Wu[2], Qingyi Tao[2], Kang Liao[1], Chen Change Loy[1]✉**
[1] S-Lab, Nanyang Technological University; [2] SenseTime Research
yi-kai.wang@outlook.com  ccloy@ntu.edu.sg
Project Page: https://yikai-wang.github.io/nvg

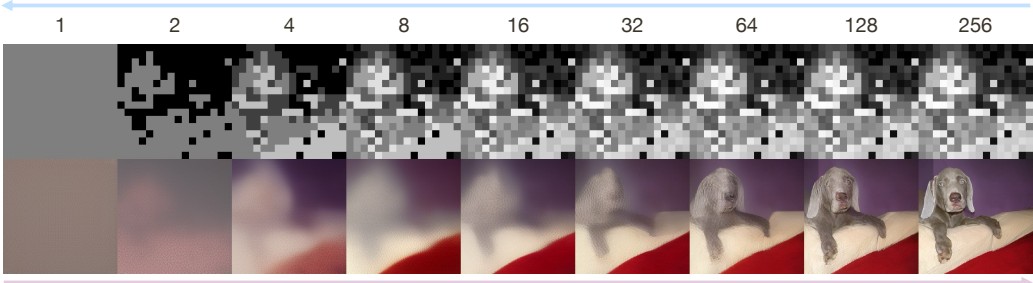

(a) Construction of the visual granularity sequence on $256^2$ image and the next visual granularity generation in the $16^2$ latent space. Top-to-bottom: Number of unique tokens, structure map, generated image.

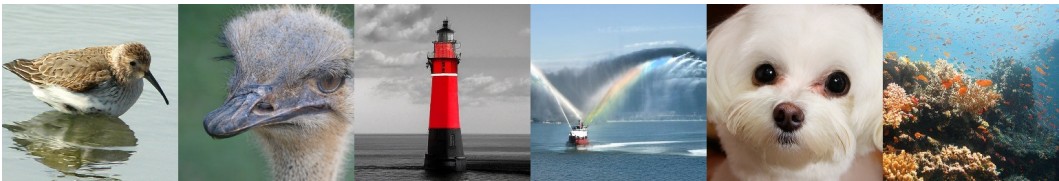

(b) Our model can generate diverse and high-fidelity images.

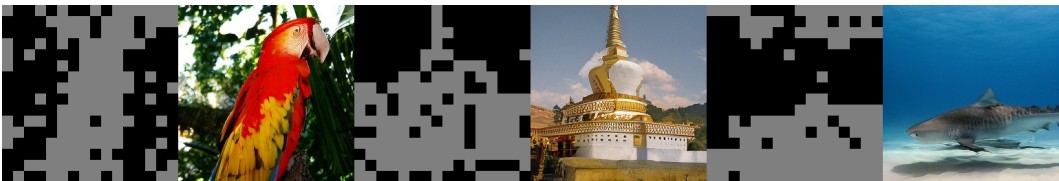

(c) The generated images align well with the generated binary structure map.

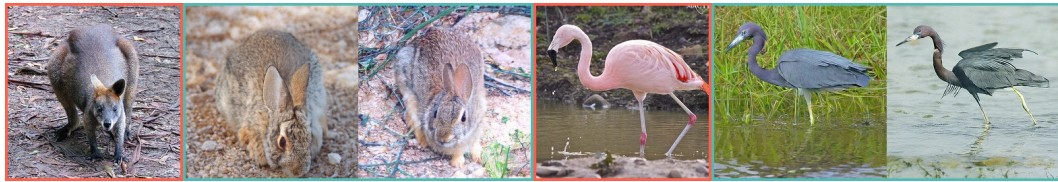

(d) We can reuse structures from reference images (wallaby, flamingo) to generate new ones (rabbits, heron).

Figure 1: We propose Next Visual Granularity (NVG) generation framework, representing images with a varying number of unique tokens, naturally forming different granularity levels. The induced structure maps reflect how these tokens are assigned across different spatial locations. The structure maps and unique tokens are iteratively generated to gradually refine the generated image.

## ABSTRACT

We propose a novel approach to image generation by decomposing an image into a structured sequence, where each element in the sequence shares the same spatial resolution but differs in the number of unique tokens used, capturing different level of visual granularity. Image generation is carried out through our newly

introduced Next Visual Granularity (NVG) generation framework, which generates a visual granularity sequence beginning from an empty image and progressively refines it, from global layout to fine details, in a structured manner. This iterative process encodes a hierarchical, layered representation that offers fine-grained control over the generation process across multiple granularity levels. We train a series of NVG models for class-conditional image generation on the ImageNet dataset and observe clear scaling behavior. Compared to the VAR series, NVG consistently outperforms it in terms of FID scores ($3.30 \rightarrow 3.03$, $2.57 \rightarrow 2.44$, $2.09 \rightarrow 2.06$). We also conduct extensive analysis to showcase the capability and potential of the NVG framework. Our code and models are released at `https://yikai-wang.github.io/nvg`.

# 1 INTRODUCTION

How do generative models understand images? Different generative models interpret images in distinct ways: Token-based models (such as autoregressive (Chen et al., 2020; Esser et al., 2021) and masked token modeling methods (Chang et al., 2022)) treat images as visual sentences, processing them similarly to how they handle language. GANs (Goodfellow et al., 2014), diffusion (Ho et al., 2020), and flow (Liu et al., 2023; Lipman et al., 2023) models see images as samples of a high-dimensional probability distribution over the raw pixel space or a learned latent space. Visual autoregressive models (Tian et al., 2024) break images down into multiple resolutions using a residual visual pyramid. In these cases, the image generation process is framed as either modeling a conditional probability distribution over previous tokens/scales or a stochastic process between the real data distribution and a random distribution.

While these approaches have led to powerful generative models, each comes with limitations in how they view and handle images: Treating an image as a sequence (like a sentence) or a pyramid often ignores the rich and complex spatial structure of images. The autoregressive methods rely on unidirectional generation, which neglects the inherent 2D spatial structure when generating early tokens, and suffer from error accumulation, also known as exposure bias (Ranzato et al., 2016). Visual autoregressive methods may mix up nearby visual information from distinct semantics and have to handle miscellaneous information. Modeling images purely as distributions often requires significant fine-tuning or extra modules (Zhang et al., 2023b) to control the generation process.

In this paper, we introduce *structured visual granularity*, representing images as structured sequences. Figure 1(a) provides an illustration of this sequence. At each stage, the image is described using different number of unique tokens in the same spatial size. This allows us to create a structure map that shows how the tokens are arranged across the latent space. The structure map naturally captures the image's granularity at different levels. This structured sequence can be used to train an image generation model that can generate images more naturally and with greater structure control.

We propose a data-driven method to build this visual granularity sequence. We use a bottom-up strategy, repeatedly clustering the most similar tokens until all tokens are merged into a single cluster that represents the entire image. Ideally, as the number of unique tokens decreases, the image structure gradually emerges: progressing from fine details to parts of the object, to objects, then to basic fore-background separation and ending with a single cluster.

Based on this sequence, we present a new Next Visual Granularity (NVG) generation process that mirrors the intuitive, coarse-to-fine progression commonly observed in art painting. Specifically, starting from an empty image, we gradually add more details in a structured manner by generating the structure map and corresponding tokens. In this way, we begin with coarse structures like foreground and background, then add object shapes, object parts, and finally fine details.

We train a series of NVG models of varying sizes on the ImageNet class-conditional image generation task. Our results reveal a clear scaling trend: performance consistently improves with larger model sizes, highlighting the scalability of our framework. Compared to other state-of-the-art image generation models, NVG achieves comparable or superior performance. In particular, when compared to VAR, our model consistently achieves better FID, IS, and recall scores across all model sizes. Figure 1(b) shows examples of diverse generated images. They align well with the generated binary

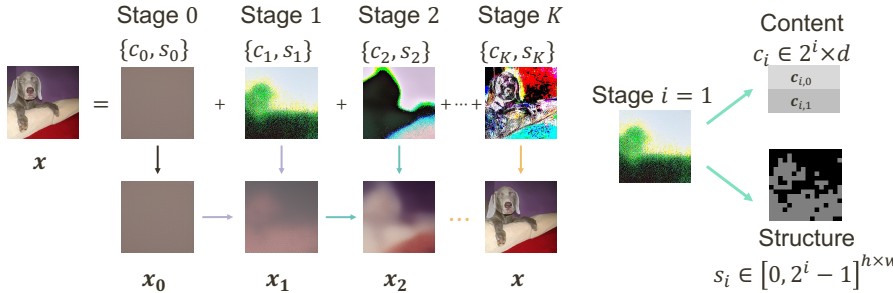

Figure 2: The relationship between current generated images in each stage $x_i$, the final generated image $x$, and the content $c_i$ and structure $s_i$ of each stage in the visual granularity sequence.

structure maps, as shown in Figure 1(c). By reusing the structure maps, NVG can transfer the structure from one image to another, generating new images with varied content, as shown in Figure 1(d).

Our approach offers several key advantages: **(1) Structured coarse-to-fine generation**: The generation process follows a natural progression of visual granularity. We introduce a structure map that explicitly controls the level of granularity in the latent space. **(2) Explicit structure control**: Each generation step controls a specific level of granularity. This structured approach enables natural control during generation itself, rather than relying on extra conditional modules trained post-hoc.

Compared to auto-regressive models, NVG uses residual modeling inspired by VAR, where each stage predicts the quantization error from the previous stages to the ground truth. This naturally reduces exposure bias. Compared to diffusion models, we introduce an explicit structure-controlled generation pipeline, making structure control an integral part of the process rather than an extra post-hoc module. Compared to VAR, our visual granularity decomposition alleviates representation ambiguity in the early stages, where a single token represents a large and semantically diverse image region. The granularity-based decomposition improves the meaning of each token, leading to better performance in both reconstruction and generation.

## 2 NEXT VISUAL GRANULARITY GENERATION

In this paper, we present the concept of *structured visual granularity*, where images are represented as structured sequences composed of varying numbers of distinct tokens distributed in the latent space at each stage. An illustration of this sequence is shown in Figure 1(a). Our method organizes the token sequence to inherently capture image granularity across multiple levels. This structured representation allows the model to generate images in a more organized and interpretable manner.

In the following, we first describe how to construct the visual granularity sequence by training a multi-granularity quantized autoencoder (Sec. 2.1). We then explain how to generate this sequence for structured image generation (Sec. 2.2).

### 2.1 VISUAL GRANULARITY SEQUENCE

**Tokenization.** We propose a multi-granularity quantized autoencoder that represents images as structured sequences across multiple levels of granularity. Specifically, an image is encoded into a latent representation $Z \in \mathbb{R}^{h \times w \times e}$, where $h$ and $w$ denote the spatial dimensions and $e$ is the channel dimension. We propose constructing a visual granularity sequence that represents a quantized latent representation. In particular, the sequence consists of *content* and *structure* pairs of multiple stages $K$, denoted as $\mathcal{T} = \{c_i, s_i\}_{i=0}^{K}$, where the contents of different stages are derived from a shared codebook $\mathcal{V} \in \mathbb{R}^{n \times e}$.

In stage $i$, the latent of size $h \times w$ is represented by $n_i$ unique tokens, dubbed as *contents*, i.e., $|c_i| = n_i, c_i \subset \mathcal{V}$. The *structure* $s_i$ is a matrix of size $h \times w$, indicating the arrangement of each token in the latent space of size $h \times w$ where the value of each position is the index in the corresponding content token, $s_i \subset \{0, 1, \ldots, n_i - 1\}^{h \times w}$. See Figure 2 for an illustration. The latent $x$ is quantized as $\sum_{i=0}^{K} \mathrm{a}(c_i, s_i)$ where $\mathrm{a}$ is the assignment operator that places the $c_i$ into the latent space according to the arrangement of $s_i$, $K$ denotes the final quantization stage.

Figure 3: We use a $K$-dim vector to encode the structure across all stages. At stage 0, all locations belong to a single cluster, so we pad the vector with all 1s. For stages $i > 0$, the embedding is inherited from the parent and extended with one extra bit (0 or 2) to distinguish between child labels.

**Structure Construction.** A structure map defines the arrangement of tokens across the latent space. To demonstrate the flexibility of our model, we propose a fully data-driven clustering approach to construct $\mathcal{T}$. Starting from the finest granularity stage, where each position is assigned a unique token, i.e., $|c_K| = hw$, we progressively group visually similar points. This can be achieved using methods like k-nearest neighbors clustering (Fix & Hodges, 1989), graph cuts (Greig et al., 1989), or linear assignment (Kuhn, 1955). As an initial exploration, we adopt a simple yet efficient greedy strategy, grouping tokens into equal-sized clusters.

Specifically, we begin by computing pairwise $\ell_2$ distances between all tokens and grouping the top-$k$ most similar tokens into a cluster. We then remove the clustered tokens from the pool and repeat the process on the remaining tokens until all tokens are assigned to a cluster. This yields the penultimate token label map $s_{K-1}$, effectively reducing the number of unique tokens by $k$. We repeat this clustering process iteratively until all tokens are merged into a single cluster, resulting in a hierarchy of multi-stage label maps $\{s_i\}_{i=0}^{K}$. We set $k = 2$, reducing the number of tokens by half in each stage. With the structure maps defined, we now derive the corresponding content tokens.

**Content Construction.** We construct the multi-stage tokens in a residual manner, forming a visual pyramid similar to VAR (Tian et al., 2024). However, unlike the spatial resizing used in VAR, our compression is guided by the induced structure map. With this design, we have a sequence of unique tokens with a count of $\{2^i\}_{i=0}^{8}$ for a latent space of size $16^2$. Algorithm 1 summarizes our construction algorithm. The detailed implementation is provided in App. A.2.

**Structure Embedding.** We introduce a compact hierarchical structure embedding for multi-stage label maps $\{s_i\}_{i=0}^{K}$ that preserves parent–child relations, distinguishes stages within a unified space, and avoids embedding cluster ID order. An example of this embedding process is shown in Figure 3. Each stage adds a bit (0 or 2) to the parent's $K$-dimensional bit-style vector, with 1 as padding; stage 0 is fully padded. This integer-valued design is RoPE-compatible and enables simple, efficient embedding construction from class and stage IDs. Padding reveals the stage, while bit patterns separate clusters. We discuss this design in App. A.3.

## 2.2 NEXT VISUAL GRANULARITY GENERATION

**Generation Pipeline.** We denote the generation of content tokens as *content generation*, and the generation of structure maps as *structure generation*. To support these, we design the Next Visual Granularity (NVG) generation framework. At each stage, we first generate the structure, followed by the corresponding content. This design allows users to optionally provide a preferred structure to guide and control the generation process. An illustration of our generation pipeline is shown in Figure 4, with the detailed algorithm provided in Algorithm 2. We use separate models for content and structure generation, respectively. We provide the architecture details in App. A.4.

### 2.2.1 STRUCTURE GENERATOR

As we explicitly represent each stage using separate channels in the structure embeddings, directly generating the whole hierarchical structure in one-step is challenging. Furthermore, the structure generator must produce a full binary cluster map of the image in first step. This task is complex and plays a crucial role in guiding successful image generation. We regard this as the "cold-start" issue.

On the other hand, compared to content generation, which typically uses a large codebook and high-dimensional channels, structure generation is simpler. It only needs to generate 8-channel embeddings, with a maximum complexity of $2^8 = 256$. Intuitively, we can use a smaller model for structure generation while applying more advanced techniques to better model the structure distribution and balance quality with efficiency.

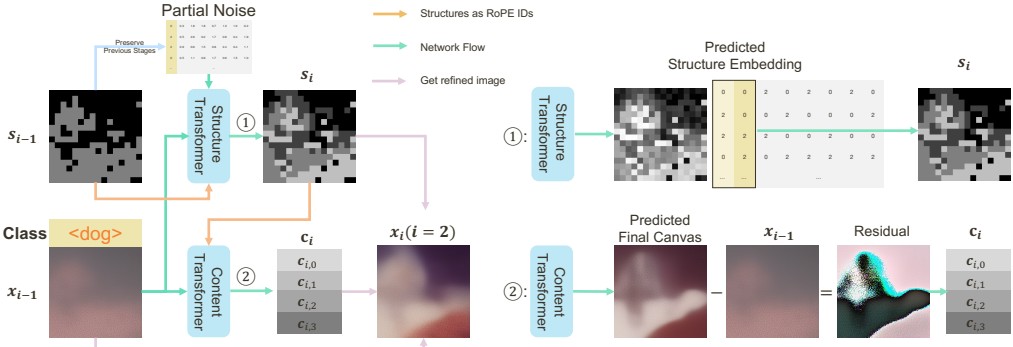

Figure 4: **Left**: Overview of our generation pipeline. At each stage, we first generate the structure and then generate the content based on that structure. Both steps are guided by the input text, the current canvas, and the current hierarchical structure. **Right**: Overview of how we obtain the target from the network predictions. (1): The structure generator predicts the overall structure embedding, from which we extract the channel for the next stage. (2): The content generator predicts the final canvas. We compute the residual between the predicted final canvas and the current canvas $x_{i-1}$, and use it to obtain the next-stage content prediction.

Based on these considerations, we propose using a lightweight rectified flow model (Liu et al., 2023; Lipman et al., 2023) for structure generation. To this end, we design a unified structure generation process that spans all stages. This approach allows training from later stages to also inspire the early stages, helping to avoid cold-start issues.

**Input.** We use the structure embeddings $s_e$ (see Figure 3 for an illustration) as the initial input. We then add the noise $\varepsilon$ to the embeddings, leading to $z_s(t) = t \cdot \varepsilon + (1 - t) \cdot s_e$. In the later stages of generation, the outputs from earlier stages are already known and fixed. Hence, for these known parts, we use ground-truth embeddings instead of noised embeddings, i.e.,

$$z_s(t)[:, 0 : i] := s_e[:, 0 : i]. \tag{1}$$

This makes our structure generation task a structure "inpainting" task. We concatenate the text and current generated image in the spatial channel as the condition for structure generation.

**Structure Prediction.** We use the $v$-prediction (Salimans & Ho, 2022) to estimate the velocity during the denoising process. During sampling, in stage 0, we start with $s_e$ filled with 1. We then use random noise as input and gradually denoise to generate the final hierarchical structure embedding $\hat{s}_e$. In the early stages, the generated full hierarchical structure embedding may be inaccurate because it relies only on the image produced so far. Therefore, at stage $i$, given the structure maps from stages 0 to $i - 1$ and the image refined from stage $i - 1$, we generate the full structure embedding, then we utilize the generated structure map of stage $i$ to update $s_e$ as:

$$s_e[:, i - 1] \leftarrow \hat{s}_e[:, i - 1]. \tag{2}$$

Note that all we need for the generated structure is to split each parent label evenly in the latent space. To this end, instead of treating the generated structures as binary splitting decisions, we interpret them as unnormalized probabilities of belonging to a particular sub-cluster. Using Gumbel-top-$k$ sampling (Kool et al., 2019), we could sample half of the locations for one sub-cluster and assign the remaining half to the other. This increases the diversity of the generated structure.

### 2.2.2 CONTENT GENERATOR

Unlike VAR (Tian et al., 2024), which models token relationships in an image-independent manner, our method captures image-specific relationships that convey richer structural information. To train effectively, the model must be aware of the content's hierarchical structure. Without this, such as when unique tokens from different stages are flattened into a 1D sequence as in VAR, the model cannot be trained effectively. We address this issue through progressive canvas refinement, where we iteratively refine the canvas by producing more structured details in different visual granularity.

Specifically, we define the *canvas* as the latent we generated so far by accumulating all previous stages, $x_i := \sum_{j=1}^{i} \mathrm{a}(c_i, s_i)$. Then the content generator is trained to directly generate the final

canvas $f_c(\boldsymbol{x}_i) \rightarrow \boldsymbol{x} := \boldsymbol{x}_K$. The difference between the final canvas (model output) and the current canvas (model input), $f_c(\boldsymbol{x}_i) - \boldsymbol{x}_i$, simulates the quantization target for the current stage's content tokens $\boldsymbol{c}_i$ during the VGS construction. An illustration of the relationship between the final canvas $\boldsymbol{x}$, current canvas $\boldsymbol{x}_i$, and current content $\boldsymbol{c}$ can be found at Figure 2. This strategy is similar to diffusion and flow models, but in contrast to denoising the image, we aim to refine the canvas iteratively.

**Inputs.** The input canvas is represented as a flattened sequence, concatenating with the text condition. The text embedding and stage embedding are summed and input to the norm layer.

**Structure-Aware RoPE.** To help the model understand the token structure, we extend RoPE (Su et al., 2024) to encode the hierarchical structure. For the attention feature dimension of 64, we split it into: $[8]$ for text/image identification, $[2] \times 8$ for structure encoding, $[20] \times 2$ for image spatial locations. In this structure-aware RoPE, for each stage, tokens within the same cluster are treated as being at the same structure position. Vice versa, if they are from different clusters, they are treated as at different structure positions. As the stages accumulate, the model knows the *entire hierarchical structure* of the tokens. The structure IDs are the structure embedding shown in Figure 3.

**Content Prediction.** We train the model to predict the final canvas $\boldsymbol{x}$. This approach unifies the training objectives across different stages and helps prevent overfitting by providing an informative supervision signal. Next, we compute the difference $f_c(\boldsymbol{x}_i) - \boldsymbol{x}_i$ between the predicted final canvas and the current input canvas. We then average the features corresponding to each unique token at this stage to obtain the feature vector for each unique token. A linear layer maps these token features to logits $\hat{\boldsymbol{c}}_i$, forming a distribution over all possible token candidates. These logit vectors serve the same role as those in (visual) autoregressive models, supporting token sampling during generation.

**Content Generator Training.** We use MSE loss to supervise the generation of the final canvas and cross-entropy loss to supervise the content prediction, as

$$\ell(\boldsymbol{x}_i) = \|\boldsymbol{x} - f_c(\boldsymbol{x}_i)\|_2^2 + \mathrm{CE}(\hat{\boldsymbol{c}}_i, \boldsymbol{c}_i). \tag{3}$$

We train the model with 10% null condition. We apply RePA (Yu et al., 2025c) at the 8-th layer.

## 3 EXPERIMENTS

We train NVG models of varying sizes for class-conditional image generation on the ImageNet dataset. We compare NVG with other models and provide further analysis of NVG. The training and sampling details and reconstruction comparisons are given in App. A.5, A.6 and App. B.1, respectively.

### 3.1 GENERATION RESULTS

**Quantitative Comparison.** We compare our NVG series with state-of-the-art image generation models in Table 1. The competing models are grouped into the following categories: GANs, diffusion models (Diff), masked auto-regressive models (Mask), standard auto-regressive models (AR), auto-regressive variants (X-AR), and VAR. We evaluate all models using standard metrics: FID, Inception Score, precision, and recall, calculated with OpenAI's evaluation tool introduced in Dhariwal & Nichol (2021). We also report each model's size, training steps, and generation steps.

Overall, our NVG models perform comparable or better than all competing methods, while requiring fewer training steps and using fewer parameters. NVG consistently outperforms VAR on FID, IS, and recall. These results highlight the strength of our framework. As we scale up the NVG model, we see consistent improvements in FID and Inception Score, showing the strong potential of our approach.

**Qualitative Visualization.** We visualize the generated images in Figure 5. **(1)** In the top rows, we display representative examples. The first unique token sets the overall color tone, while the first binary structure map defines the initial layout. As the generation progresses, the structure map becomes more detailed, guiding finer aspects of the layout. Meanwhile, the image itself is refined step-by-step, starting with object shapes, then object parts, and finally visual details. **(2)** Our clustering algorithm is based on feature similarity, so the binary structure map often reflects a rough separation between foreground and background. However, the generator can interpret this map flexibly. This is evident in the middle row: while the generator generally follows the structure, it may merge separated regions to form one object (e.g., the chimpanzee's chest). It may also split a single region, such as the foreground, into multiple distinct objects (e.g., several ostriches). **(3)** The

Table 1: Generation performance on class-conditional ImageNet $256 \times 256$.

| Type | Model | FID(↓) | IS(↑) | Pre(↑) | Rec(↑) | #Para | #Train† | #Step |
|---|---|---|---|---|---|---|---|---|
| GAN | BigGAN (Brock et al., 2019) | 6.95 | 224.5 | 0.89 | 0.38 | 112M | - | 1 |
| | GigaGAN (Kang et al., 2023) | 3.45 | 225.5 | 0.84 | 0.61 | 569M | 920K | 1 |
| | StyleGan-XL (Sauer et al., 2022) | 2.30 | 265.1 | 0.78 | 0.53 | 166M | - | 1 |
| Diff | CDM (Ho et al., 2022) | 4.88 | 158.7 | — | — | — | 2.1M | 8100 |
| | LDM-4-G (Rombach et al., 2022) | 3.60 | 247.7 | — | — | 400M | 178K | 250 |
| | DiT-XL/2 (Peebles & Xie, 2023) | 2.27 | 278.2 | 0.83 | 0.57 | 675M | 7M | 250 |
| | SiT-X (Ma et al., 2024) | 2.06 | 270.3 | 0.82 | 0.59 | 675M | 7M | 250 |
| Mask | MaskGIT (Chang et al., 2022) | 6.18 | 182.1 | 0.80 | 0.51 | 227M | 300e | 8 |
| | RCG (cond.) (Li et al., 2023) | 3.49 | 215.5 | — | — | 502M | 200e+800e | 20 |
| | TiTok-S-128 (Yu et al., 2024b) | 1.97 | 281.8 | — | — | 287M | 300e | 64 |
| | MAGVIT-v2 (Yu et al., 2024a) | 1.78 | 319.4 | — | — | 307M | 1080e | 64 |
| | MAR-H (Li et al., 2024) | 1.55 | 303.7 | 0.81 | 0.62 | 943M | 800e | 256 |
| AR | VQGAN (Esser et al., 2021) | 15.78 | 74.3 | — | — | 1.4B | 2.4M | 256 |
| | RQTran. (Lee et al., 2022) | 7.55 | 134.0 | — | — | 3.8B | - | 68 |
| | ViTVQ (Yu et al., 2022) | 4.17 | 175.1 | — | — | 1.7B | 450K | 1024 |
| | LlamaGen-L (Sun et al., 2024) | 3.07 | 256.1 | 0.83 | 0.52 | 343M | 300e | 576 |
| | LlamaGen-XXL (Sun et al., 2024) | 2.34 | 253.9 | 0.80 | 0.59 | 1.4B | 300e | 576 |
| | Open-MAGVIT2-XL (Luo et al., 2024) | 2.33 | 271.8 | 0.84 | 0.54 | 1.5B | 300e∼350e | 256 |
| | IBQ-XL (Shi et al., 2024) | 2.14 | 279.0 | 0.83 | 0.56 | 1.1B | 300e∼450e | 256 |
| | IBQ-XXL (Shi et al., 2024) | 2.05 | 286.7 | 0.83 | 0.57 | 2.1B | 300e∼450e | 256 |
| X-AR | DART-FM (Gu et al., 2025) | 3.82 | 263.8 | — | — | 820M | 500K | 16 |
| | SAR-XL (Liu et al., 2024) | 2.76 | 273.8 | 0.84 | 0.55 | 893M | 200e | 256 |
| | RandAR-L (Pang et al., 2025) | 2.55 | 288.8 | 0.81 | 0.58 | 343M | 300e | 88 |
| | RandAR-XXL (Pang et al., 2025) | 2.15 | 322.0 | 0.79 | 0.62 | 1.4B | 300e | 88 |
| | D-AR-XL (Gao & Shou, 2026) | 2.09 | 298.4 | 0.79 | 0.62 | 775M | 300e | 256 |
| | EAR-H (Shao et al., 2025) | 1.97 | 289.6 | 0.81 | 0.59 | 937M | 800e | 64 |
| | CausalFusion-XL (Deng et al., 2024) | 1.77 | 282.3 | 0.82 | 0.61 | 676M | 800e | 250 |
| VAR | VAR-$d16$ (Tian et al., 2024) | 3.30 | 274.4 | 0.84 | 0.51 | 310M | 200e | 10 |
| | VAR-$d20$ (Tian et al., 2024) | 2.57 | 302.6 | 0.83 | 0.56 | 600M | 250e | 10 |
| | VAR-$d24$ (Tian et al., 2024) | 2.09 | 312.9 | 0.82 | 0.59 | 1.0B | 350e | 10 |
| Ours* | NVG-$d16$ (255M+64M) | 3.03 | 279.2 | 0.82 | 0.54 | 320M | 200e | 9 |
| | NVG-$d20$ (497M+125M) | 2.44 | 310.4 | 0.80 | 0.60 | 622M | 250e | 9 |
| | NVG-$d24$ (856M+215M) | 2.06 | 317.0 | 0.79 | 0.61 | 1.1B | 350e | 9 |

* We treat content and structure generation as one step per stage. Counting them separately gives $9 * 1 + 7 * n$ steps in total. In this paper, we use $n = 25$ and we leave generating structures with one-step flow models as future work.

† This column shows training iterations/epochs. K: thousand iterations, M: million, e: epochs. With batch size 768, 100e $\approx$ 167K. The training iterations for NVG are 333K, 444K, and 519K, respectively.

bottom rows highlight the diversity and quality of our results. We provide a qualitative comparison with other methods in App. B.3.

## 3.2 FURTHER ANALYSIS

We provide ablation study in App. B.2, generation variation analysis in App. B.4, and complexity analysis in App. B.5.

**Structure-Guided Generation.** One key advantage of our framework is we can use an explicit structure map to guide generation. We test our framework with very simple binary structure maps based on basic geometric shapes, such as circles placed in different positions or a rectangle. The results are shown in Figure 6. We also provide reference structure-guided generation results at the bottom of Figure 6. The model follows the given structure maps closely. Because the structure maps are continuous, the generated background tends to be simple. The model fills in the foreground according to the provided class label, though in some cases it interprets the structure map differently but still in a reasonable way. In practice, one could use segmentation maps or semantic layouts to create structure maps, enabling more detailed and controlled generation. This demonstrates the benefit of our framework: it supports flexible control right out of the box without additional training.

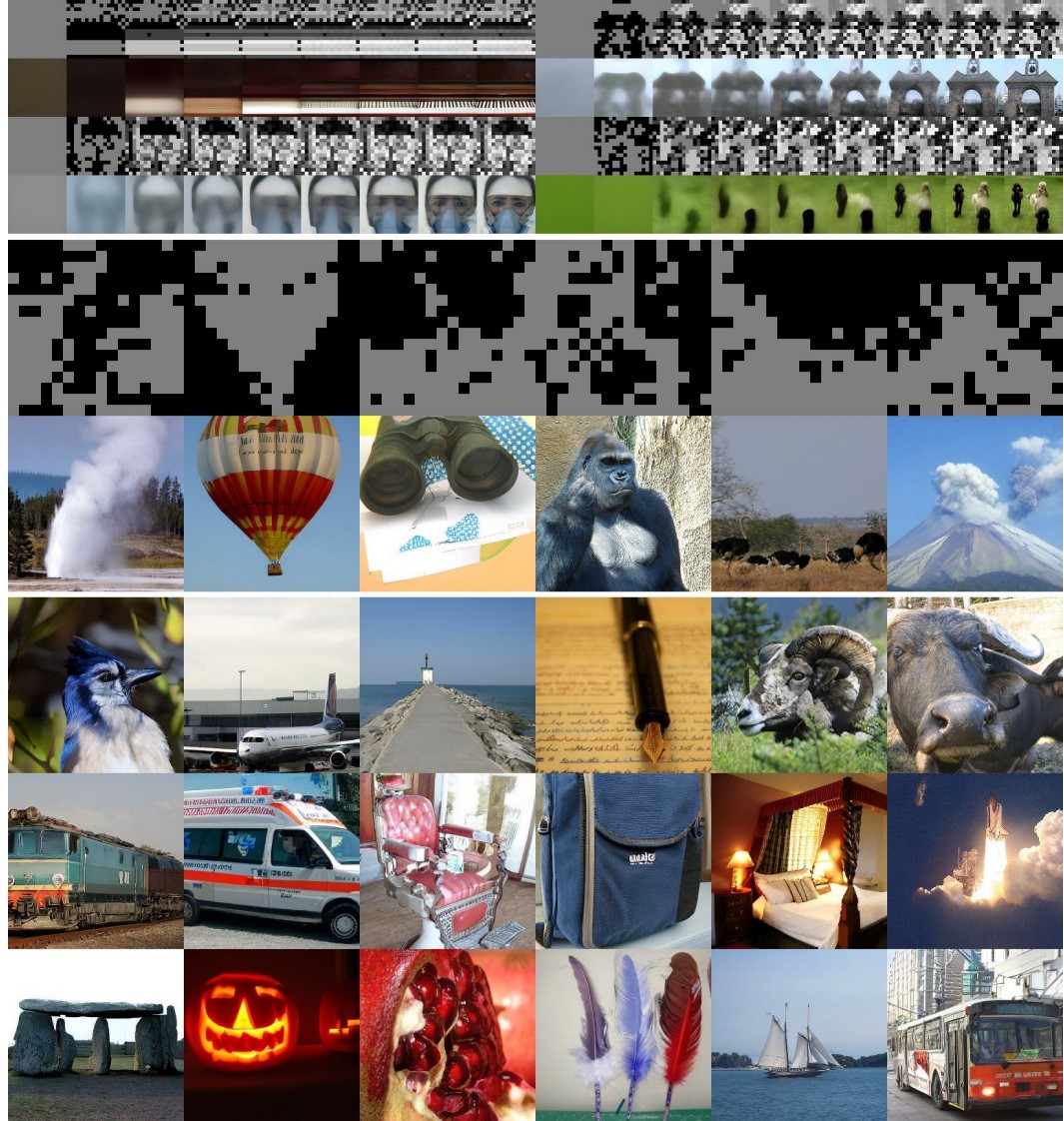

Figure 5: Visualization of generated images. **Top**: We show several representative examples to illustrate the iterative generation process. **Middle**: The generated binary structure maps align well with the final images. **Bottom**: Our NVG-$d24$ model can generate diverse and high-quality images.

**Extreme Cases Analysis.** In this section, we examine how NVG behaves in several extreme cases. Representative examples are shown in Figure 7.

(a) *Unclear structures*: In some situations, the generated structures appear unclear or less interpretable. NVG handles these cases as follows: (1) Rough object shapes: When realistic object boundaries do not align perfectly with the generated structures, NVG first produces a coarse but plausible structure and then progressively refines it into more accurate shapes (first row of Figure 7a). (2) Uninterpretable structures but coherent images: Sometimes the structures look ambiguous to humans, but the resulting images still show a clear spatial layout (second row of Figure 7a). We argue that these structures remain meaningful to the generator. (3) Extreme cases: When the object has a color very similar to the background or is extremely small, the content generator first adjusts the overall layout and textures in the early stages. Once more flexibility is allowed in later stages, precise and realistic objects emerge and are quickly refined (last row of Figure 7a).

(b) *Multi-objects cases*: Images containing multiple objects are particularly challenging. NVG typically handles them in three ways: (1) Small objects: When objects are very small, they are often merged into a single large region. NVG then refines this region in the final stages to produce clear

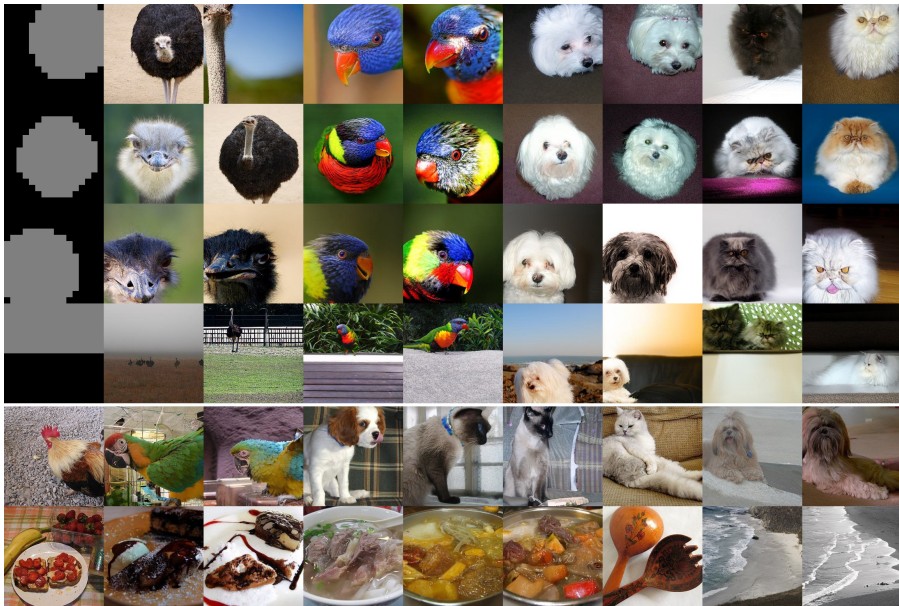

Figure 6: Visualization of structure-guided generation results. **Top**: Each row shows generated images based on the given geometric binary structure map. **Bottom**: Each group of three images includes one reference image and two generated images that follow its structure.

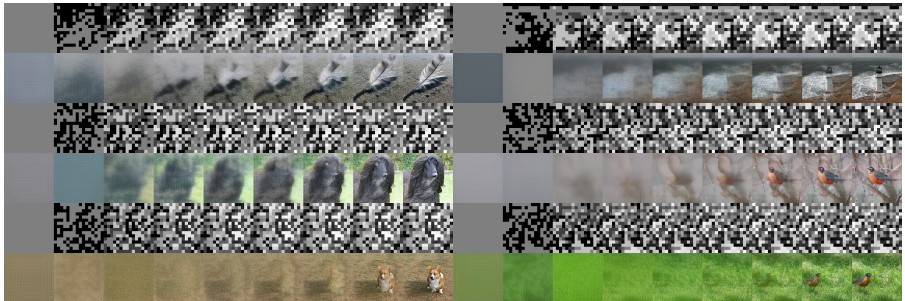

(a) Our NVG can still generate realistic images even when the structure maps are rough (first row), unclear to humans (second row), or in extreme imbalance cases (third row).

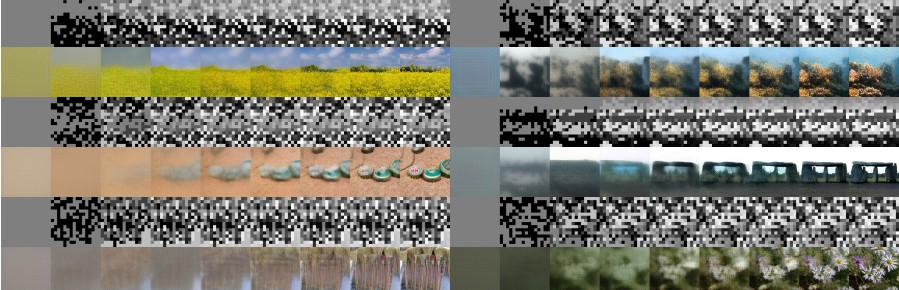

(b) Our NVG can generate multiple objects of various sizes like small ones (first row) as well as moderate ones (second row), and even those with structures that are unclear to humans (third row).

Figure 7: NVG can handle several extreme cases where equal-size, value-based clustering or the generated structures are not ideal.

object boundaries (see the first row of Figure 7b). (2) Moderate-sized objects: When objects have moderate size, their structural patterns emerge early and are refined in the middle stages. This leads to clearer object separation, as shown in both the structures and images in the second row of Figure 7b.

(3) Unclear structure maps: In extreme cases where the structure map is unclear to humans, NVG still captures the overall scene layout, as seen in the early refined images. As refinement continues, the model gradually separates individual objects and enhances textures (last row of Figure 7b).

Overall, NVG shows strong robustness in handling complex scenes and imperfect structures.

## 4 RELATED WORK

The underlying modeling of current image generation models can be grouped into holistic, fragmented, and cascaded image modeling processes. We provide a discussion of these methods in App. C. Most existing approaches lack explicit modeling of the structural composition inherent in images. We aim to address this gap by prompting structure-awareness as a fundamental aspect in image generation.

## 5 CONCLUSION AND DISCUSSIONS

This work advances image generation by explicitly modeling hierarchical visual structure, addressing a key limitation of existing approaches that often treat images as flat, unstructured data. By decomposing images into sequences of increasing granularity and guiding generation through structure-aware mechanisms, our framework not only improves fidelity but also opens new possibilities for structure-controlled generation. The strong empirical performance demonstrates that our NVG framework can scale effectively with model size, suggesting a promising path toward more controllable generative systems. Beyond the benchmarks, the ability to separate and iteratively refine structure and content offers practical advantages in domains such as design, scientific visualization, and any scenario where structure and hierarchy in the generation process are essential.

**Limitations and Future Work.** In this paper, we focus on verifying the feasibility of our generation framework and developing an effective learning paradigm. We also provide fair comparisons with competitors to better demonstrate the capability of our NVG models. Our study centers on standard class-conditional image generation using the ImageNet benchmark dataset. For future work, there are several exciting directions to explore in visual granularity-based generation:

*Better Structure Design:* In this paper, we derive the structure directly from data. (1) Depending on the task, the structure map can be designed to include expert knowledge about the image, such as multi-level segmentation, frequency decomposition, or resolution scales. (2) When the structure is learned from data, more advanced algorithms can be used to split the image into more interpretable regions, such as more accurate clustering methods or flexible clustering approaches that explicitly handle imbalanced regions in the image. We believe all these variants can be effectively trained within our proposed framework.

*Region-Aware Generation:* Our approach enables direct generation using visual granularity sequences defined by domain-specific annotations. These controls are not just conditions. They define the structure the model follows. Our multi-stage setup also allows fine-grained control over specific regions and generation stages.

*Physical-Aware Video Generation:* Structured image regions can be tracked over time, enabling video generation that is more coherent and physically realistic. By tracking the evolution of each region over time, we can enforce structural constraints and physical laws during generation, rather than applying post-hoc supervision to the generated frames.

*Hierarchical Spatial Reasoning:* The pioneering work Spatial Reasoning Models (Wewer et al., 2025) explores visual spatial reasoning using a patch-wise diffusion process with predicted patch generation order such that the reasoning is enabled when generating unobserved patches conditioned on observed patches. Our method can also be used to generate a structured global-to-local divide-and-conquer reasoning chain for spatial reasoning.

**Reproducibility Statement.** We have made every effort to ensure our work is reproducible. The main paper provides a complete description of the framework and pipeline, while the appendix includes detailed information on the implementation, architecture, and optimization. The full training and inference source code and models are available in `https://yikai-wang.github.io/nvg`.

**Acknowledgments.** This study is supported under the RIE2020 Industry Alignment Fund Industry Collaboration Projects (IAF-ICP) Funding Initiative, as well as cash and in-kind contribution from the industry partner(s). It is also partially supported by NVIDIA Academic Grant. We thank Hanze Dong, Ziang Zhou, and Junqiu Yu for thoughtful discussions.

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

## A  METHOD

### A.1  ALGORITHM

We present our construction and generation algorithms in Algorithm 1 and Algorithm 2, respectively.

---

**Algorithm 1:** VGS Construction

1 **Inputs:** Encoded latent $\boldsymbol{Z}$, $K = 8$, token down factor $m = 2$, latent size $L = 16 \times 16$;
2 **Initialize:** $\boldsymbol{s}_K = \text{range}(L)$, $\boldsymbol{c}_0$ from Eq. (4) ;
3 **for** $k = K-1, \cdots, 0$ **do**
4     $\boldsymbol{Z}_k, \boldsymbol{s}_k = \text{Cluster}(\boldsymbol{Z}_{k+1}, \boldsymbol{s}_{k+1}, m)$;
5 **for** $k = 1, \cdots, K$ **do**
6     Reorganize $\boldsymbol{s}_k$;
7     Calculate $\boldsymbol{R}_k$ from Eq. (5);
8     Calculate $\boldsymbol{c}_k$ from Eq. (4);
9 **Return:** VGS $\mathcal{T} = \{\boldsymbol{c}_i, \boldsymbol{s}_i\}_{i=1}^K$;

---

**Algorithm 2:** VGS Generation

1 **Inputs:** Class condition $txt$, $\boldsymbol{\varepsilon}_s \sim \mathcal{N}(0,1)$;
2 **Initialize:** $\boldsymbol{x} = 0$, $\boldsymbol{s}_e = [1]^{L \times K}$;
3 **for** $k = 0, \cdots, K$ **do**
4     **if** $0 < k < K$ **then**
5        $\boldsymbol{s}_e \leftarrow f_s(txt, \boldsymbol{x}, \boldsymbol{s}_e, \boldsymbol{\varepsilon}_s, k)$;
6     # No need to predict $s_{K-1} \rightarrow s_K$
7     Get $\boldsymbol{s}_k$ from Eq. (7);
8     $\boldsymbol{c}_k = f_c(txt, \boldsymbol{x}, \boldsymbol{s}_e, k)$;
9     $\boldsymbol{x} = \boldsymbol{x} + \boldsymbol{s}_k(\boldsymbol{x}_k)$;
10 **Return:** $\mathcal{T} = \{\boldsymbol{c}_i, \boldsymbol{s}_i\}_{i=1}^K$ or $\boldsymbol{x}$;

---

### A.2  CONTENT CONSTRUCTION ALGORITHM

In stage-0, we initialize the quantization target $\boldsymbol{R}_0$ as the latent $\boldsymbol{Z}$ and quantize it to obtain content token $\boldsymbol{c}_0$ as:

$$\boldsymbol{c}_i = Q(\text{Avg}(\boldsymbol{R_i}, \boldsymbol{s}_i)), \text{ where } Q(\boldsymbol{z}) := \min_{\mathcal{V}_j} \|\mathcal{V}_j - \boldsymbol{z}\|_2. \tag{4}$$

Avg is an average operator applied within the locations of the same cluster ID. After getting the tokens of each stage, we calculate the quantization error via

$$\boldsymbol{R}_i := \boldsymbol{Z} - \sum_{j=0}^{i-1} \boldsymbol{R}_j - \phi_i \circ \text{a}(\boldsymbol{c}_i, \boldsymbol{s}_i), \tag{5}$$

where $\text{a}(\cdot)$ is an assign operator that assigns the unique tokens to the original latent space following the structure map $\boldsymbol{s}_i$, and $\phi_i$ is a single conv refiner layer as in VAR (Tian et al., 2024). The $\boldsymbol{R}_i$ serves as the target for quantization in the next stage.

### A.3  STRUCTURE EMBEDDING CONSTRUCTION

Structure maps play a crucial role in our generation framework as they determine the layout of details at different granularity levels. To this end, We propose a compact, hierarchical representation to encode the multi-stage label maps $\{\boldsymbol{s}_i\}_{i=0}^K$ at each spatial location. The structure embedding is designed to satisfy the following criteria: (1) Parent-child relationships across stages must be preserved; (2) Embeddings from different stages should remain distinguishable within a unified embedding space; (3) The ordering of cluster IDs should not be incorporated in the embedding, as they serve only to distinguish between clusters. In practice, we also expect the implementation of the mapping from cluster ID and stage ID to the embedding is simple and efficient.

We define the structure embedding as a bit-style vector in the most-to-least significant bit order, with bit values 0 and 2 and padding value 1 as a midpoint. An example of this embedding process is shown in Figure 3. We use a $K$-dimensional vector to encode the structure across all stages. At stage 0, since all locations belong to a single cluster, we pad the vector with all 1s. For stages $i > 0$, the embedding is inherited from the parent and extended with one additional bit (either 0 or 2) to distinguish between child labels. All values are non-negative integers for easy integration with RoPE IDs (Su et al., 2024). We apply this structure embedding in both the content and structure generators in our framework.

During construction, we reindex the structure map sequence such that class $j$ in stage $i$ becomes classes $2j$ and $2j + 1$ in stage $i + 1$. With this implementation, we can derive the embedding by converting the structure label $s$ into bits using the formula:

$$\boldsymbol{s}_e(s, i)_j = 2(\lfloor s/2^{i-1-j} \rfloor \bmod 2)1_{j<i} + 1_{j \geq i}, \ \forall 0 \leq j < K, \forall \, 0 \leq i \leq K, \forall s \in [0, 2^i). \quad (6)$$

The original structure class ID $s$ can be directly recovered from the structure embedding via

$$s = \sum_{j=0}^{i-1} \boldsymbol{s}_e[j] \times 2^{i-j-2}. \quad (7)$$

Our structure embedding strictly follows the intended design criteria. Specifically: (1) Each stage is represented by one channel. Child embeddings are derived from their parent embeddings by adding a new bit to distinguish between child labels, effectively preserving the parent-child relationship. (2) The number of padding channels directly indicates the stage of the structure embedding, making the hierarchical structure clearly visible across channels. (3) The bit-style embedding clearly separates different clusters within a unified space, without relying on the order of cluster IDs.

### A.4 ARCHITECTURE

**Content Generator** We follow FLUX (Black-Forest-Labs, 2024) to use the self-attention block with parallel linear layers (Dehghani et al., 2023) as our basic building block. We follow VAR for network configuration, setting network width $w$, attention heads $h$ and dropout rate $dr$ with model depth $d$ as:

$$w = 64d, \quad h = d, \quad dr = 0.1 \cdot d/24. \quad (8)$$

The major number of parameters in the backbone grows with model depth according to:

$$N(d) = d \cdot (\underbrace{7w^2}_{\text{qkv and mlp\_in}} + \underbrace{5w^2}_{\text{proj and mlp\_out}} + \underbrace{3w^2}_{\text{modulation}}) = 15dw^2 = 61,440d^3. \quad (9)$$

This is $5/6$ the size of VAR since we adopt a simpler modulation design inspired by FLUX.

**Structure Generator** Since the structure embeddings have a lower dimension (8 v.s. 32) and fewer unique tokens (up to 256 v.s. 4096) than the content embeddings, learning the structure is comparatively easier. Therefore, we reduce the model width to $1/2$ that of the content generator while keeping other architectural aspects the same, leading to

$$w = 32d, \quad h = d/2, \quad dr = 0.1 \cdot d/24, \quad N(d) = 15dw^2 = 15,360d^3. \quad (10)$$

This results in a small model that is $1/4$ the size of the content generator. When combined, our full generator has only about $1/24$ more parameters than VAR. We also adopt the structure-aware RoPE in the structure generator, where the first part is extended to text/image/structure identification.

### A.5 TRAINING SETUP

We train the auto-encoder using the same loss functions as Open-MAGVIT2 (Luo et al., 2024), but replace the original discriminator with DINO (Zhang et al., 2023a), following the setup used in VAR. To improve codebook initialization, we apply IBQ (Shi et al., 2024) during the first epoch as a warm-up. We set the base channel size to 128 instead of 160 which is used in VAR. The codebook has a size of $4096 \times 32$. We train the autoencoder for 100 epochs with a batch size of 256 and a learning rate of $1e-4$ with cosine decay to 0. The generator is trained with a base learning rate of $1e-4$ and a batch size of 256, scaled linearly based on the actual batch size. We use the WSD learning rate schedule (Hu et al., 2024): We linearly increase the learning rate during the first 1,000 steps as a warmup, then keep it constant for most of the training, and finally decrease it linearly to $1/10$ of the original rate by the end of training. We train NVG-$d16$ for 200 epochs, NVG-$d20$ for 250 epochs, and NVG-$d24$ for 350 epochs to make a fair comparison with VAR. For NVG-$d16$ and NVG-$d20$, the learning rate starts to decay after 80% of the training epochs. For NVG-$d24$, we found that the model's behavior stabilizes between 120 and 200 epochs, so we begin decaying the learning rate at epoch 200 to avoid wasting more computing resources. We train our models on ImageNet (Deng et al., 2009). The generation step for structure generator is set as $n = 25$.

Table 2: Reconstruction performance on ImageNet validation dataset. Results of VAR are reproduced, while other competitors are from IBQ. LPIPS are calculated via VGG (Simonyan & Zisserman, 2015).

| Tokenizer | #Tokens | Ratio | Codebook | rFID($\downarrow$) | LPIPS($\downarrow$) | Usage ($\uparrow$) |
|---|---|---|---|---|---|---|
| VQ-GAN (Esser et al., 2021) | $16 \times 16$ | 16 | 1,024 | 7.94 | - | 44% |
| SG-VQGAN (Rombach et al., 2022) | $16 \times 16$ | 16 | 16,384 | 5.15 | - | - |
| VQGAN-LC (Zhu et al., 2024) | $16 \times 16$ | 16 | 100,000 | 2.62 | 0.2212 | 99% |
| MaskGIT (Chang et al., 2022) | $16 \times 16$ | 16 | 1,024 | 2.28 | - | - |
| LlamaGen (Sun et al., 2024) | $16 \times 16$ | 16 | 16,384 | 2.19 | 0.2281 | 97% |
| Open-MAGVIT2 (Luo et al., 2024) | $16 \times 16$ | 16 | 262,144 | 1.17 | 0.2038 | 100% |
| IBQ (Shi et al., 2024) | $16 \times 16$ | 16 | 262,144 | 1.00 | 0.2030 | 84% |
| VAR (Tian et al., 2024) | 680 | 16 | 4,096 | 1.06 | 0.1863 | 100% |
| NVG | $511^{*}$ | 16 | 4,096 | **0.74** | 0.1875 | 100% |

[*] We define #Tokens as the number of unique tokens. If consider the number of total tokens, our model uses $256 \times 9$ tokens. In comparison, VAR uses $256 \times 10$ tokens, showing that our tokenizer achieves better quantization with fewer tokens.

Table 3: Ablation on NVG-$d12$. Models are trained for 40 epochs, tested with a constant CFG scale of 1.5 for both content and structure generators without setting TOP_K or TOP_P.

| Content Input | Structure Input | Structure RoPE | FID($\downarrow$) | IS($\uparrow$) | Precision($\uparrow$) | Recall($\uparrow$) |
|---|---|---|---|---|---|---|
| $x + t\varepsilon$ | Pure Noise | ✓ | 38.94 | 44.0 | 0.48 | 0.53 |
| $x$ | Pure Noise | ✓ | 37.89 | 44.6 | 0.50 | 0.53 |
| $x$ | Partial Noise | ✗ | 39.03 | 43.5 | 0.49 | 0.52 |
| $x$ | Partial Noise | ✓ | **37.59** | **45.1** | **0.50** | **0.53** |

## A.6 SAMPLING METHOD

For content generation, as we are learning the token distribution, we might follow autoregressive approaches to sample the tokens to improve diversity. This sampling approach is effective in the early stages, where the model focuses on the overall contents of the image. However, because we use a residual learning approach, the model is trained to predict the difference between the current canvas and the final canvas, i.e., fix the error. Hence in later stages, the process becomes less about sampling and more about accurately correcting errors. Inspired by this, we propose control the available candidates by reducing the TOP_P parameter from $100\%$ (the full codebook) down to $50\%$ (the most confident candidates) in a logarithmic rate. This strikes a balance between promoting diversity early on and ensuring high fidelity in the later stages. We follow previous methods by linearly increasing the CFG scale: from 1 to 2.5 during structure generation, and from 1 to 3.5 during content generation. This helps strengthen guidance in the later stages of generation.

## B EXPERIMENT

### B.1 RECONSTRUCTION COMPARISON

We compare our tokenizer with state-of-the-art methods in Table 2. Our tokenizer achieves better rFID (Heusel et al., 2017) and comparable LPIPS (Zhang et al., 2018) scores, while using a much smaller codebook and maintaining a high utilization rate. Our tokenizer significantly outperforms VAR's in reconstruction quality with less number of unique tokens, showing the advantage of our granularity-based approach over scale-based tokenization. Furthermore, in the first stage, VAR's codebook utilization rate is $25.39\%$, while ours is $68.55\%$, indicating a more balanced codebook.

### B.2 ABLATION STUDY

We conduct ablation studies on NVG-$d12$. The results are summarized in Table 3.

*(a) Content Inputs*: We evaluate three types of content inputs: (1) Current Canvas ($x$): Similar to autoregressive modeling, where the current canvas is used directly; (2) Noised Canvas ($x + t\varepsilon$): Adds Gaussian noise, following the approach of EDM (Karras et al., 2022), trying to incorporate more randomness during generation; (3) Variance-Preserving Noised Canvas ($(1 - t)x + t\varepsilon$): A

linear interpolation between the canvas and noise, inspired by rectified flow models (Liu et al., 2023; Lipman et al., 2023). The noise level $t$ is linearly scheduled as $t = 1 - (i - 1)/9$ for stage $i$.

During training, the variance-preserving version performed significantly worse in predicting tokens compared to other approaches. As a result, it is considered ineffective, and we terminate its training. These results suggest that using an autoregressive approach is more effective for generating the VGS, indicating that treating content generation as conditional modeling is better.

*(b) Structure Inputs*: We test two variants: (1) Pure Noise: Uses standard Gaussian noise for the entire structure input. (2) Partial Noise: Replaces known parts of the Gaussian noise with ground-truth values for known stages, mimicking a structure inpainting task. The results show that preserving known stages leads to better performance.

*(c) Structure-Aware RoPE* : We compare models with and without structure-aware RoPE to evaluate its impact. When only spatial-aware RoPE is used without structure-aware RoPE, the model struggles to capture structural relationships between tokens, resulting in weaker generation quality.

*(d) Content Prediction*: We also tried directly predicting the next content without predicting the final canvas. However, this method starts to overfit after about 25 epochs and is therefore considered ineffective and we terminate its training. This further highlights the advantage of providing richer supervision through the final canvas.

## B.3 QUALITATIVE COMPARISON

In this section, we show random examples generated by our method and several state-of-the-art baselines, including diffusion models (SiT-X (Ma et al., 2024)), autoregressive models (IBQ-XXL (Shi et al., 2024)), and VAR (Tian et al., 2024), as shown in Figure 9 and Figure 10. These results show that NVG achieves improved or comparable image quality and diversity compared with these SOTA methods.

## B.4 STAGE-WISE GENERATION VARIATION

We analyze generation variation in different stages by fixing the structure and content from earlier stages and generate the remaining stages. The results are shown in Figure 8.

*(a) In-domain controlled generation*: In the left column, we guide the generation using the in-domain class "standard poodle". The provided structure and content align with this class label.

(1) The first unique token has a strong influence on the main semantics of the image, such as the presence of a dog and a grassy background. *The first binary structure map determines the overall layout.* When only the structure map is fixed, the generated images show dogs with similar poses but different colors and backgrounds. *When the first unique token is fixed, the overall semantic content and color tones stay consistent across generations.* This doesn't mean the first token directly specifies colors or specific objects, but it provides a strong prior that guides the generator towards that direction. We can confirm this by using different class guidance in the right column. When we change the class condition, the same fixed content and structure can lead to noticeably different outputs, as the model "interprets" them differently. A similar effect is seen with changes to the structure map.

(2) As more stages are fixed, the variation in the generated images decreases. In stages 2–3, layout details are fixed. This reduces variation in the dog's pose and how the background is arranged. These constraints come from both the fixed content and structure. If only the structure is fixed, the generator still creates multiple plausible variations that match the structure. In stages 4–5, detailed aspects like the dog's head orientation become fixed. Content tokens clearly encode information such as the dog facing forward. If only structure is fixed, the interpretation of the generator varies, but the results remain consistent and will not change when fixing more stages. The final stages refine smaller details, such as the texture of the dog's fur in the face or the color of flowers in the background. These results show a clear trend: *each stage controls a different level of visual information, from general layout and semantics to fine-grained appearance details.*

*(b) Out-of-domain controlled generation*: In the right column, we guide the generation process using distinct class labels such as "Siamese cat" and "Indian elephant". These results demonstrate the model's ability to produce diverse outputs and correct errors.

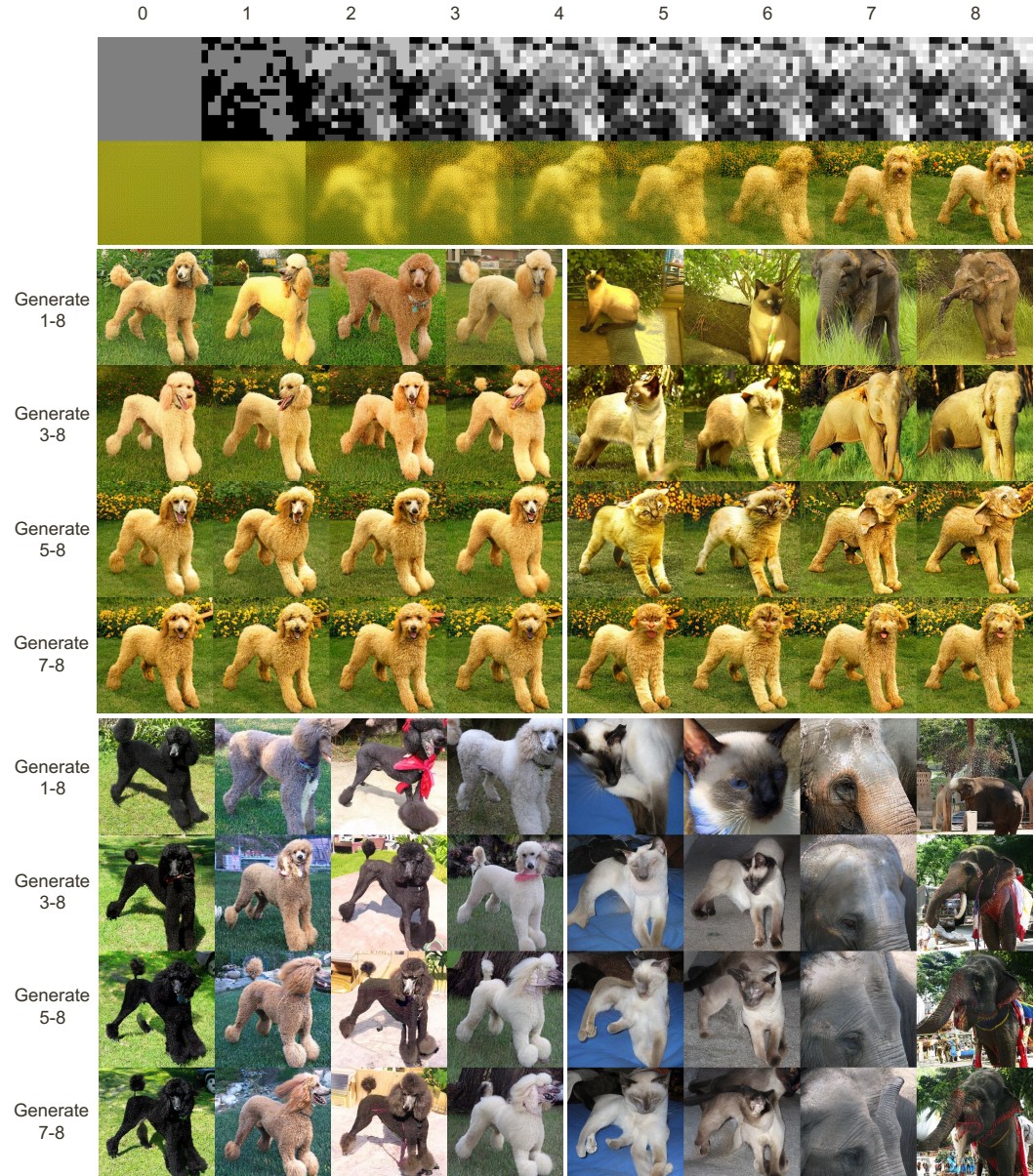

Figure 8: Visualization of stage-wise controlled generation results. **Top** row shows how the reference image is reconstructed at each stage. **Middle** row shows generated images where both the content and structure from previous stages are preserved. **Bottom** row shows generated images where only the structure from previous stages is preserved. The **left** column uses the class "standard poodle" as guidance, which closely resembles the reference image. The **right** column uses the classes "Siamese cat" and "Indian elephant" as guidance, representing out-of-distribution (OOD) examples. In the case of "Generate $i$-8", the structure of $i$-stage is provided by the reference image.

(1) When both content and layout are fixed in the early stages, the overall color and layouts are also fixed. However, if the target class differs from the reference image, the model struggles to maintain layout control when only the first stage is fixed. This suggests that different classes tend to follow different structural patterns. The generator tends to interpret the layout based on what it has learned during training. For example, cats often appear in indoor scenes. However, if we fix the first unique token, the generated image changes to an outdoor scene with grass in the background.

(2) The structure maps can be interpreted in multiple valid ways. Even when the structure map comes from a single image, the generator can still produce diverse results. But once the content is fixed

Table 4: Comparsion of inference cost.

| Model | #Para | Time Cost (s) | Memory Cost (MiB) |
|---|---|---|---|
| SiT-X | 675M | 1.403360 | 4099 |
| IBQ-XL | 1.1B | 5.848799 | 5563 |
| VAR-d16 (w/ KV cache) | 310M | 0.115876 | 2521 |
| NVG-d16 (w/o structure generation) | 255M | 0.073220 | 1563 |
| NVG-d16 | 320M | 0.723339 | 1687 |
| VAR-d20 (w/ KV cache) | 600M | 0.137775 | 3817 |
| NVG-d20 (w/o structure generation) | 497M | 0.084362 | 2253 |
| NVG-d20 | 622M | 0.841313 | 2495 |
| VAR-d24 (w/ KV cache) | 1.0B | 0.159784 | 5539 |
| NVG-d24 (w/o structure generation) | 856M | 0.112932 | 3001 |
| NVG-d24 | 1.1B | 1.122031 | 3471 |

along with the structure, this variety disappears. Interestingly, even when the first three stages of a dog image are fixed, our model can still generate an image of a different class. *This shows that our framework has strong error-correction ability.* This is a major advantage over autoregressive models, which cannot revise what they have already generated.

## B.5    Complexity Analysis

In this section, we analyze the time and memory usage of NVG and compare it with state-of-the-art generation models in Table 4. All models are evaluated under the same hardware, software, data type, acceleration techniques, and CFG settings. The VAR series is further accelerated using KV cache, following their official implementation. The reported runtime and memory usage cover the full generation process in the latent space as well as the final decoding stage. Because NVG generates structure maps with unique tokens while VAR can be regarded as using a fixed structure map, we also include a variant of NVG that uses a fixed structure map for a more fine-grained comparison.

(1) *Content generation.* Thanks to our efficient generation sequence (9 steps v.s. 10 in VAR) and our improved architecture, NVG achieves noticeably faster runtime than VAR. VAR consumes more memory mainly due to the KV cache it uses for acceleration, while NVG does not rely on this mechanism.

(2) *Structure generation.* Introducing the structure generator, currently implemented with a flow-matching model, increases the runtime. Because the structure model is small, the additional memory overhead remains limited.

(3) *Overall comparison.* Although structure generation makes our method slower than the VAR series, it is still much faster than standard diffusion models (SiT-X) and autoregressive models (IBQ-XL). Notably, our method also uses much less memory than current state-of-the-art models. Speeding up the structure generation step is an important direction for future work, and we believe it is achievable given the recent rapid progress in few-step flow-matching models.

## C    Related Works

**Holistic Image Modeling.**    These models treat the image as a unified high-dimensional distribution in pixel/latent space, aiming to generate the entire image simultaneously. To approximate this distribution, models typically start from a simple prior, such as a Gaussian distribution, and transform it into the image distribution. Prominent modeling approaches include generative adversarial networks (Goodfellow et al., 2014), diffusion models (Ho et al., 2020), and flow-based methods (Liu et al., 2023; Lipman et al., 2023). GANs generally perform one-step image generation, whereas diffusion models and flow-based methods often rely on multi-step generation processes. In particular, diffusion and flow-based models are by design require hundreds or thousands of generation steps. They reduce the number of generation steps during inference via specific sampling algorithms (Song et al., 2021). These models are known for producing high-fidelity, high-quality generation results (Kang et al., 2023; Rombach et al., 2022; Esser et al., 2024) and can be conditioned on auxiliary inputs through extra module (Zhang et al., 2023b) or light-weight fine-tuning (Hu et al.,

2022). However, the holistic nature of the generation process introduces challenges for precise control. They usually need to train additional components to control the generation process.

**Fragmented Image Modeling.**    These models conceptualize an image as a set of discrete, often non-overlapping patches, analogous to a 2D sentence of visual words. This approach is inspired by autoregressive modeling in natural language processing (Bengio et al., 2003; Brown et al., 2020). The model learns to predict the distribution of unknown patches conditioned on observed ones. The generation order can follow a raster-scan pattern (autoregressive) (Chen et al., 2020; Esser et al., 2021) or a model-defined sequential rule (non-autoregressive) (Chang et al., 2022; Pang et al., 2025; Yu et al., 2025b). The conceptual alignment with text modeling facilitates integration into multi-modal architectures. These models also need hundreds or even thousands of steps to produce results. To speed up, non-autoregressive models often generate multiple tokens in one step, while autoregressive models mainly rely on engineering techniques like the KV-cache.

**Cascaded Image Modeling.**    These models adopt a hierarchical, coarse-to-fine generation strategy. Typically, a low-resolution image is generated first, followed by subsequent upsampling and refinement to achieve high-resolution output (Razavi et al., 2019; Ho et al., 2022). Recent work starts decomposing images at the same spatial resolution. The pioneer work on visual autoregressive models (Tian et al., 2024) uses spatial compression to control the density of visual information in different stages. Other kinds of decompositions like the number of tokens (Bachmann et al., 2025; Gao & Shou, 2026) or frequency filter (Yu et al., 2025a) have also been explored. Li et al. (2025) explores generating image in a divide-and-conquer way. Cascaded models benefit from faster generation due to their few-step design, making them generally more efficient than diffusion or autoregressive models without further exploration on sampling algorithms or engineering techniques.

# D    STATEMENTS

**Use of Large Language Models.**    We follow ICLR's policy and use LLMs only as general-purpose assistive tools. In our work, they were used to polish the writing in some paragraphs and to suggest improvements for code. They were not involved in research ideation, experimental design, or the development of core technical contributions.

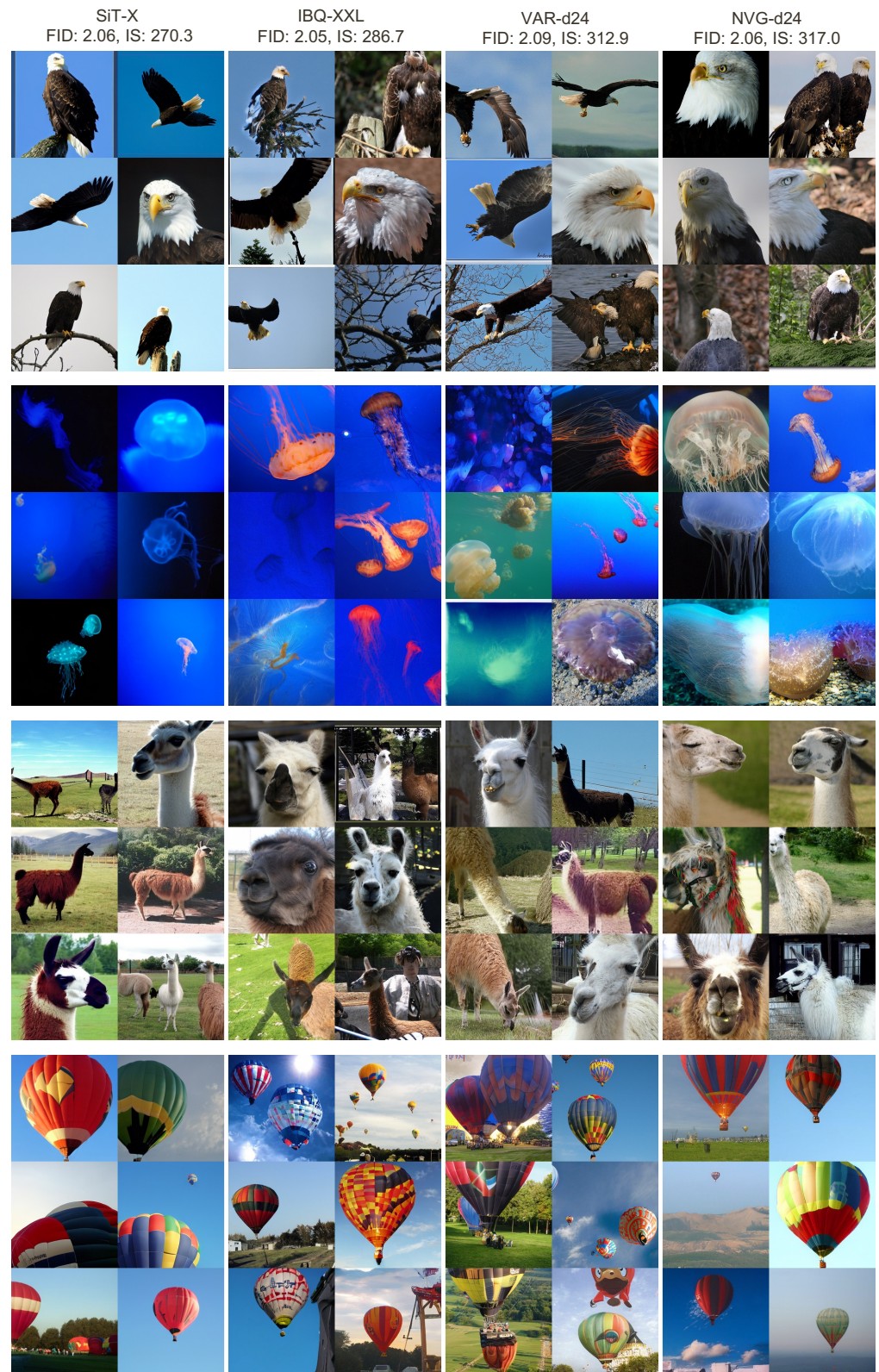

Figure 9: ualitative comparison with state-of-the-art diffusion (SiT-X (Ma et al., 2024)), auto-regressive (IBQ-XXL (Shi et al., 2024)) and VAR (Tian et al., 2024) models.

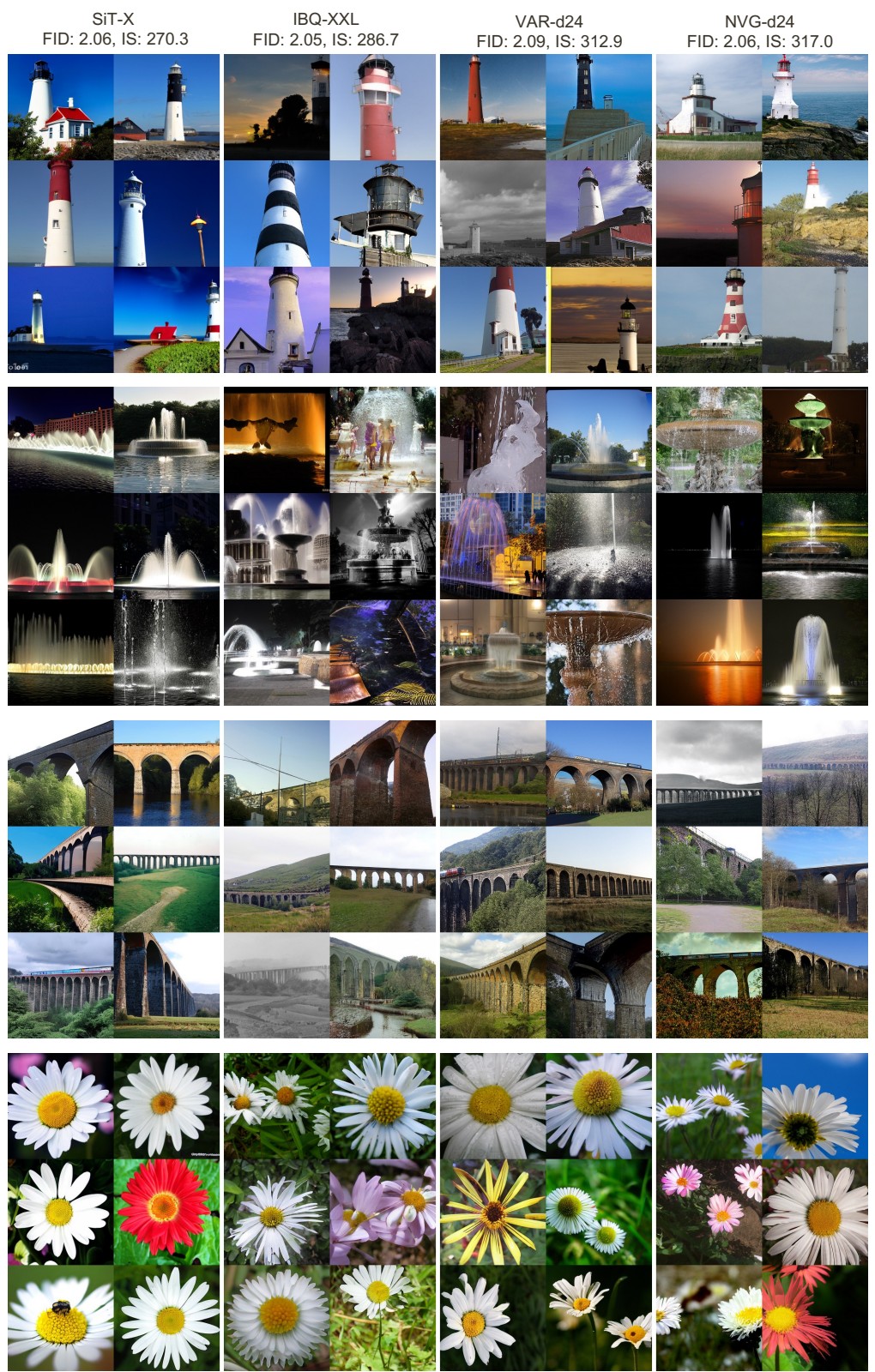

Figure 10: More qualitative comparison with state-of-the-art diffusion (SiT-X (Ma et al., 2024)), auto-regressive (IBQ-XXL (Shi et al., 2024)) and VAR (Tian et al., 2024) models.

