# OpenReview forum: "Next Visual Granularity Generation"
_ICLR.cc/2026/Conference — ICLR 2026 Poster_

### Official Review · Reviewer_82ro · 2025-10-21

**Soundness:** 3
**Presentation:** 3
**Contribution:** 3
**Rating:** 6
**Confidence:** 4

**Summary:**

This paper proposes Next Visual Granularity Generation (NVG), a hierarchical autoregressive image generation paradigm that progressively increases the token number (the “visual granularity”) with spatial resolution fixed in multiple stages. Specifically, during the inference process of each stage, a structure map (granularity structure information) is predicted with a diffusion model and then AR model is leveraged to transfer the stage-specific image to a fake final image with this structure, producing new stage residual content. On class-conditional ImageNet, NVG shows consistent but modest FID/IS improvements over VAR-style next-resolution autoregression and presents appealing spatial controllability via structure map guidance.

**Strengths:**

1. Novel Concept. Using the diversity token and explicit structure map to decouple and capture the different-level visual granularity, achieving good explainational exploration and controllable generation for AR.

2. Good Methodological Techniques. The integration of diffusion model and AR with structure-aware positional embedding for explicit structure map generation and granularity visual content producing is neat and well-motivated.

3. Empirical Experiments. On ImageNet class-conditional generation, NVG exhibits clear scaling trends with model size. Visualization of the structure-map controllability demonstrates a practical interface for layout control.

**Weaknesses:**

1. Structure construction assumptions. In this paper, the structure map plays a core role during the generation process. Since the structure map is constructed with a hierarchical clustering algorithm as ground-truth labels, such construction method may misalign with semantic boundaries in some situations (e.g., crowded/occluded object scenes) and then leads to failure generation. Robustness to such cases is unclear. Additionally, how about the generation results with a noisy or bad structure map during inference?


2. Limited improvement magnitude. Gains over strong VAR baselines are small (e.g., FID reductions on the order of ~0.2–0.3 absolute). The authors may further explore the advantages of structure-map controllability for other applications.

3. System metrics are underreported. The two-stage pipeline (structure and content generation) and multi-stage decoding likely incur additional costs, but this paper lacks memory footprints and time cost curves.

**Questions:**

1. Sensitivity about stage number. How sensitive are results to the number of granularity stages?

2. Fake Final Canvas Prediction. Why does the model not produce the residual directly (like a diffusion process that predicts denoising noise)?

---

> ### Author Response · Authors · 2025-11-20
> **Response to Reviewer 82ro**
>
> We sincerely appreciate your time and effort in reviewing our paper.
>
> **W1: Complex or bad structures?**
>
> Thank you for raising this issue. Our model can generate complex scenes, and even when the generated structures are not perfect, they still capture the overall layout and produce realistic images. We visualize and discuss several representative cases in Figure 7.
> 1. **Unclear structures**:
> NVG handles these cases as follows:
> - Rough object shapes: When realistic object boundaries do not align perfectly with the generated structures, NVG first produces a coarse but plausible structure and then progressively refines it into more accurate shapes (first row of Figure 7(a)).
> - Uninterpretable structures but coherent images:
> Sometimes the structures look ambiguous to humans, but the resulting images still show a clear spatial layout (second row of Figure 7(a)).
> We argue that these structures remain meaningful to the generator.
> - Extreme cases: When the object has a color very similar to the background or is extremely small, the content generator first adjusts the overall layout and textures in the early stages.
> Once more flexibility is allowed in later stages, precise and realistic objects emerge and are quickly refined (last row of Figure 7(a)).
>
> 2. **Multi-objects cases**:
> NVG handles multi-object case in three typical ways:
> - Small objects:
> When objects are very small, they are often merged into a single large region.
> NVG then refines this region in the final stages to produce clear object boundaries (see the first row of Figure 7(b)).
> - Moderate-sized objects:
> When objects have moderate size, their structural patterns emerge early and are refined in the middle stages.
> This leads to clearer object separation, as shown in both the structures and images in the second row of Figure 7(b).
> - Unclear structure maps:
> In extreme cases where the structure map is unclear to humans, NVG still captures the overall scene layout, as seen in the early refined images.
> As refinement continues, the model gradually separates individual objects and enhances textures (last row of Figure 7(b)).
>
> Overall, NVG demonstrates strong robustness in handling complex scenes and imperfect structural generation.
>
> **W2: Limited improvement magnitude?**
>
> We would like to highlight the comparison with VAR is designed to demonstrate the superiority of our structure-induced sequence design. We construct our model with approximately the same number of total parameters with VAR. Notably, our content generator is only 5/6 the size of VAR’s, meaning we use a smaller model for content generation. Under this setup, the structure generator is even smaller, 1/4 size of our content generator. Since VAR can be seen as using fixed, spatially-related structure maps, this configuration ensures that any performance improvement comes from our proposed structure-induced image sequence. Therefore, we consider this improvement meaningful for a fundamentally new generation framework.
>
> We appreciate the suggestion to explore the advantages of structure-map controllability in other applications. We discussed several promising directions in Sec. 5 and plan to investigate them in future work. In this paper, our main focus is on proposing and validating a fundamentally new image generation framework.

---

> > ### Author Response · Authors · 2025-11-20
> > **Response to Reviewer 82ro (Cont.)**
> >
> > **W3: Memory footprint and time cost report?**
> >
> > Thank you for this question. First please note that the decoding from latent space to image space is a single decoding step, not a “multi-stage decoding”.
> > We report memory and time comparisons with VAR.
> >
> > | Model                                  | #Para | Time Cost (s) | Memory Cost (MiB) |
> > |----------------------------------------|-------|---------------|--------------------|
> > | **SiT-X**                              | 675M  | 1.403360      | 4099               |
> > | **IBQ-XL**                             | 1.1B  | 5.848799      | 5563               |
> > | **VAR-d16 (w/ KV cache)**              | 310M  | 0.115876      | 2521               |
> > | **NVG-d16 (w/o structure generation)** | 255M  | 0.073220      | 1563               |
> > | **NVG-d16**                            | 320M  | 0.723339      | 1687               |
> > | **VAR-d20 (w/ KV cache)**              | 600M  | 0.137775      | 3817               |
> > | **NVG-d20 (w/o structure generation)** | 497M  | 0.084362      | 2253               |
> > | **NVG-d20**                            | 622M  | 0.841313      | 2495               |
> > | **VAR-d24 (w/ KV cache)**              | 1.0B  | 0.159784      | 5539               |
> > | **NVG-d24 (w/o structure generation)** | 856M  | 0.112932      | 3001               |
> > | **NVG-d24**                            | 1.1B  | 1.122031      | 3471               |
> >
> > All models are evaluated under the same hardware, software, data type, acceleration techniques, and CFG settings. The VAR series is further accelerated using a KV cache, following their official implementation. The reported runtime and memory usage cover the full generation process in the latent space as well as the final decoding stage. Because NVG generates structure maps with unique tokens while VAR uses a fixed structure map, we also include a variant of NVG that uses a fixed structure map for a more fine-grained comparison with VAR.
> > 1. *Content generation*.
> > Thanks to our efficient generation sequence (9 steps versus 10 in VAR) and our improved architecture, NVG achieves noticeably faster runtime than VAR. VAR consumes more memory mainly due to the KV cache it uses for acceleration, while NVG does not rely on this mechanism.
> >
> > 2. *Structure generation*.
> > Introducing the structure generator, currently implemented with a flow-matching model, increases the runtime. Because the structure model is small, the additional memory overhead remains limited.
> >
> > 3. *Overall comparison*. Although structure generation makes our method slower than the VAR series, it is still much faster than standard diffusion models (SiT-X) and autoregressive models (IBQ-XL).
> > Notably, our method also uses much less memory than current state-of-the-art models.
> > Speeding up the structure generation step is an important direction for future work, and we believe it is achievable given the recent rapid progress in few-step flow-matching models.
> >
> > **Q1: Sensitivity about stage number?**
> >
> > Due to resource and time limits during the rebuttal period, we were unable to systematically test other choices of stage numbers. Similar to VAR, our method decomposes an image into a fixed sequence. Because both the autoencoder and the generator must use the same stage numbers during training and inference, testing other stage numbers would require full retraining.
> > In early experiments, we trained an autoencoder with an 8× downsampling size (our paper uses 16×) and 1/4 clustering tokens (our paper uses 1/2), leading to a sequence of [1,4,16,64,256,1024]. This model achieved a reconstruction FID of 0.40, which is comparable to SOTA methods. To ensure consistency with the VAR setup, we adopted our current configuration. Still, these early results suggest that the number of stages is not very sensitive.
> >
> > **Q2: Why does the model not produce the residual directly?**
> >
> > We already tested this variant in Sec. B.2(d), where the model directly predicts the next stage’s content. The results were poor, confirming that predicting the final canvas provides richer supervision and leads to better outcomes.

---

### Official Review · Reviewer_aGuw · 2025-10-24

**Soundness:** 3
**Presentation:** 3
**Contribution:** 3
**Rating:** 6
**Confidence:** 4

**Summary:**

The paper introduces Next Visual Granularity (NVG), a generative framework that models image synthesis as a coarse-to-fine process guided by hierarchical structure maps. It constructs these maps through a greedy clustering of token embeddings, progressively merging similar regions to capture visual granularity from details to global structure. NVG alternates between structure generation and content generation at each stage, offering interpretable and controllable image synthesis. Experiments on ImageNet show competitive or superior performance to VAR across FID, IS, and recall, demonstrating improved structure awareness and scalability.

**Strengths:**

1.	The proposed structure prediction follows an intuitive and interpretable coarse-to-fine process, aligning well with how humans perceive visual composition.
2.	The framework effectively separates structure and content generation, enabling controllable and interpretable image synthesis.
3.	The model demonstrates strong scalability and competitive performance across metrics such as FID and Inception Score compared to state-of-the-art baselines.

**Weaknesses:**

1.	Semantic Misalignment in Clustering
NVG constructs hierarchical structures via greedy clustering based on token embedding similarity.
However, embedding-space similarity does not always reflect semantic consistency; tokens with similar color or texture may be grouped together even if they belong to different objects, leading to inaccurate granularity segmentation.
2.	Increased Computational Cost
Compared with VAR, NVG requires two sequential generation steps, structure and content at each stage.
This design improves interpretability but likely increases inference time and computational complexity, offsetting VAR’s efficiency advantage over traditional AR and diffusion models. It would be helpful if the paper reported the actual inference-time difference between NVG and VAR at a similar number of generation steps.

**Questions:**

1.	Since NVG heavily relies on the accuracy of the structure maps at each stage, would errors in early-stage structure generation propagate and amplify through subsequent stages, leading to degraded image quality?
2.	In cases involving multiple objects within a scene, how effectively can the structure map distinguish between different foreground objects? Does the current clustering-based construction capture such object-level boundaries?
3.	Although Table 1 reports a similar number of generation steps as VAR, NVG introduces separate structure and content generation at each stage. Does the observed performance gain come at the cost of increased computational load or slower inference speed in practice?

---

> ### Author Response · Authors · 2025-11-20
> **Response to Reviewer aGuw**
>
> We sincerely appreciate your time and effort in reviewing our paper.
>
> **W1: Semantic Misalignment in Clustering?**
>
> We agree that our clustering method is not perfect, but our model can handle these semantic misalignment cases. We discuss this in detail in our response to Q1.
>
> In this paper, our main focus is to propose a general framework that can effectively generate the structural sequence of the image. Within this framework, better clustering rules (e.g., based on segmentation or depth) can easily be explored. As an exploratory work, we show that even intuitive clustering can be effectively trained and leads to reasonable structural representations that generalize well, demonstrating the effectiveness and generalizability of our framework. We add the discussion of this point in the limitation and future work section (L516-L522).
>
> **W2&Q3: Time Complexity?**
>
> Thank you for this question. We provide the actual inference time comparison with VAR.
>
> | Model                                  | #Para | Time Cost (s) | Memory Cost (MiB) |
> |----------------------------------------|-------|---------------|--------------------|
> | **SiT-X**                              | 675M  | 1.403360      | 4099               |
> | **IBQ-XL**                             | 1.1B  | 5.848799      | 5563               |
> | **VAR-d16 (w/ KV cache)**              | 310M  | 0.115876      | 2521               |
> | **NVG-d16 (w/o structure generation)** | 255M  | 0.073220      | 1563               |
> | **NVG-d16**                            | 320M  | 0.723339      | 1687               |
> | **VAR-d20 (w/ KV cache)**              | 600M  | 0.137775      | 3817               |
> | **NVG-d20 (w/o structure generation)** | 497M  | 0.084362      | 2253               |
> | **NVG-d20**                            | 622M  | 0.841313      | 2495               |
> | **VAR-d24 (w/ KV cache)**              | 1.0B  | 0.159784      | 5539               |
> | **NVG-d24 (w/o structure generation)** | 856M  | 0.112932      | 3001               |
> | **NVG-d24**                            | 1.1B  | 1.122031      | 3471               |
>
> In this table, all models are evaluated under the same hardware, software, data type, acceleration techniques, and CFG settings. The VAR series is further accelerated using KV cache, following their official implementation. The reported runtime and memory usage cover the full generation process in the latent space as well as the final decoding stage. Because NVG generates structure maps with unique tokens while VAR can be regarded as using a fixed structure map, we also include a variant of NVG that uses a fixed structure map for a more fine-grained comparison.
> 1. *Content generation*.
> Thanks to our efficient generation sequence (9 steps versus 10 in VAR) and our improved architecture, NVG achieves noticeably faster runtime than VAR. VAR consumes more memory mainly due to the KV cache it uses for acceleration, while NVG does not rely on this mechanism. These results indicate a less computational load in the content generation part of our NVG.
>
> 2. *Structure generation*.
> Introducing the structure generator, currently implemented with a flow-matching model, increases the runtime. Because the structure model is small, the additional memory overhead remains limited.
>
> 3. *Overall comparison*. Although structure generation makes our method slower than the VAR series, it is still much faster than standard diffusion models (SiT-X) and autoregressive models (IBQ-XL).
> Notably, our method also uses much less memory than current state-of-the-art models.
> Speeding up the structure generation step is an important direction for future work, and we believe it is achievable given the recent rapid progress in few-step flow-matching models.
>
> Furthermore, the introduced structural sequence provides an inherent fine-grained control over the image structures in multi-level granularities, which all previous generation models cannot provide, indicating an unique property of our generation framework in addition to the performance gain.

---

> > ### Author Response · Authors · 2025-11-20
> > **Response to Reviewer aGuw (Cont.)**
> >
> > **Q1: Influence of error in early structures?**
> >
> > Thank you for the question. The answer is mostly no. Since structures are derived from the images themselves and are sometimes imperfect, the model already learns to handle these cases. We visualize several representative cases in Figure 7(a). NVG handles these cases as follows:
> > 1. Rough object shapes: When realistic object boundaries do not align perfectly with the generated structures, NVG first produces a coarse but plausible structure and then progressively refines it into more accurate shapes (first row).
> > 2. Uninterpretable structures but coherent images:
> > Sometimes the structures look ambiguous to humans, but the resulting images still show a clear spatial layout (second row).
> > We argue that these structures remain meaningful to the generator.
> > 3. Extreme cases: When the object has a color very similar to the background or is extremely small, the content generator first adjusts the overall layout and textures in the early stages.
> > Once more flexibility is allowed in later stages, precise and realistic objects emerge and are quickly refined (last row).
> >
> > Overall, NVG demonstrates strong robustness in handling complex scenes and imperfect structural generation.
> >
> > **Q2: multi-object case?**
> >
> > Thank you for this question. Our NVG could handle multi-object cases. We visualize several representative cases in Figure 7(b).
> > NVG handles multi-object case in three typical ways:
> > 1. Small objects:
> > When objects are very small, they are often merged into a single large region.
> > NVG then refines this region in the final stages to produce clear object boundaries (see the first row).
> > 2. Moderate-sized objects:
> > When objects have moderate size, their structural patterns emerge early and are refined in the middle stages.
> > This leads to clearer object separation, as shown in both the structures and images in the second row.
> > 3. Unclear structure maps:
> > In extreme cases where the structure map is unclear to humans, NVG still captures the overall scene layout, as seen in the early refined images.
> > As refinement continues, the model gradually separates individual objects and enhances textures (last row).

---

### Official Review · Reviewer_bnRk · 2025-10-29

**Soundness:** 3
**Presentation:** 2
**Contribution:** 3
**Rating:** 6
**Confidence:** 4

**Summary:**

This work proposes a new scheme for image generation, namely next visual granularity generation (NVG). In each granularity, the image’s latent is represented by the different binarized tokens. Each digit of the binarized token represents the grouping information of the pixel under each granularity levels. During the generation, the model start with the most coarse granularity level, where all pixels are separated into two groups. After each prediction stage, the number of the groups is doubled. This process is repeated until that all pixels have a distinct group. The model is trained in two parts. One model predicts the structure and the other predicts the content. In each prediction stage, the model first predicts structure and then predicts the content of each pixel.

**Strengths:**

* The reviewer likes the general idea of generating image from coarse-to-fine structure and content.
* The model shows its scalability, indicating it can be a potential model for foundational generative models.
* By leveraging the structure from reference image, the proposed method can conduct guided generation from reference image.

**Weaknesses:**

My major concern are related to some detail in the method and figures, the current manuscript makes the reviewer being confused.
* From the Fig 2, it seems that the models (structure and content) are predicting the residual results (that will be aggregated to final result). However, in Fig 4, the output of the structure/content model are final structure/canvas (and ln 269 also mentioned that content generator directly generated final canvas). And the “Residual” is the difference between the final canvas and the input. This is quite confusing, why two models outputs final structure and canvas, and why the residual is computed after the final canvas is obtained. In addition, there are different colored lines used in the Fig 4, the reviewer would like author to provide clear instructions on what does these colors mean.
* From the ln 239, it seems the model is conducting a flow-matching-like prediction. the reviewer has two questions here:
    * why does the author adopt this type of input?
    * is this strategy applied on each stage? If it is, does it mean it takes a flow-matching-like prediction for each step?
* The reviewer is quite not understand the ln 263-265, “To train effectively, …, the model cannot be trained effectively”. What is the thing that hampers the model training?
* The reviewer does not find the candidate pools of the content tokens. Since ln 289 mentions that “forming a distribution over all possible token candidates”, the reviewer feels like there is a candidate pool like text generation. However, the image generation is usually a continuous task and the tokens are usually continuous. I wonder that are the tokens here. Also, in the ln 269-270, the author mentioned the x-f(x) is the quantized target. I wonder what quantization there is and why use the difference as target.

**Questions:**

Please see my weaknesses

---

> ### Author Response · Authors · 2025-11-20
> **Response to Reviewer bnRk**
>
> We sincerely appreciate your time and effort in reviewing our paper.
>
> **W1: Clarification on Fig 2 and Fig 4?**
>
> Thanks for the question.
>
> Figure 2 illustrates how the sequence is defined, without including any model, as described in L152–157 and L179–184. For this visual granularity sequence, each stage measures the difference between the previously quantized result and the continuous latent representation.This difference is then decomposed into content and structure components, which serve as the generation targets for that stage.
>
> To generate these components, we design our generation framework as shown in Figure 4. Please note that the generation targets at each stage are not necessarily the direct outputs of the network. Specifically, we explain the design choices for the outputs of the content generator (L260–272, L282–288) and the structure generator (L244–256). We summarize them as follows:
> 1. Training the models to predict the final structure/canvas unifies the training targets across stages, enabling more effective model training.
> 2. In early stages, input information is coarse, so the generated final structure/canvas may be imperfect. Instead of directly using these outputs, we derive predictions for the next stage from them. This design is inspired by diffusion models, where x0​ can be estimated at each step, but iterative denoising is still applied for better results.
> 3. We indeed tried directly predicting next-stage tokens (Sec. B.2(d)), but this caused severe overfitting, so we consider it ineffective.
>
> Thank you for noting the color in Fig. 4. We have revised this figure to improve clarity and added explanations for the lines. Please let us know if anything remains unclear.
>
> **W2. Structure Generator?**
>
> *Input*: As discussed in L235–L242 and ablated in Tab. 3, we insert already known structures so that the model can better generate the remaining parts. The experiments verify the effectiveness of this design.
>
> *Flow-matching-like prediction for each step*: Yes. As discussed in L204–212, we use a flow-matching model to ensure the structure sequence generation can be effectively produced. To demonstrate feasibility, we used a strong generation model with a small network (1/4 the size of the content generator) to balance quality with efficiency. Because our goal in this work is to demonstrate the feasibility of the framework, we did not explore newer few-step flow-matching models, although they could be easily substituted for our current standard flow-matching model in future work.
>
> **W3. Why the VAR-style approach fails in our case?**
>
> Thank you for this question. The VAR-style model was our first attempt for the content generator, but it was difficult to train effectively.
>
> For example, at stage 2, VAR-style model takes 2 unique tokens as input and outputs 4 unique tokens for the next stage.
> We tried two variants:
>
> (a) Vanilla VAR: inputs 2 tokens and outputs 4 tokens.
>
> (b) VAR + structure: inputs 2 tokens with their structure map, and outputs 4 tokens.
>
> Variant (a) lacks overall structure and produces random results. Variant (b) struggles to learn token–structure relationships and child-parent arrangements (for instance, in the structure sequence, a class [A] is divided into two classes [B] and [C] in the new stage. For the output tokens, both the arrangement of [BC] and [CB] are valid, but the model is hard to produce this set-style outputs).
>
> Because of these issues, we conclude that the model’s input and output should remain 2D rather than flattened. Based on this insight, we further explored other variants in Table 3 (Sec. B.2).
>
> **W4. (a) Quantized Tokenizer?**
>
> Thank you for the question. As stated in L149–151 and Sec. A.5, we use a discrete tokenizer, the same as in VAR. The codebook is a learnable fixed-size codebook of 4096 tokens (L796), shared by all stages (L151). The quantization target and candidate pool are both from this codebook.
>
> **(b) Difference as quantization target?**
>
>  Thank you for the question. As stated in L180, we follow VAR and use a visual residual pyramid. At each stage, we use more tokens to model the quantization error from the previous stage. Hence the difference is our quantization target. Please refer to Sec. A.2 in our original paper  for a detailed explanation of how we construct the content tokens.

---

> > ### Comment · Reviewer_bnRk · 2025-11-26
> > **Follow up questions**
> >
> > The reviewer appreciates author's response. I have several follow up questions.
> >
> > My question about the structure generation is not asking for justification of the effectiveness of the design, but about the reason (or say motivation) to adopt this design. The flow-matching is designed for the continuous space while the structure embedding seems to be a discrete target (just like what Fig 3 shows). Therefore, the reviewer is confused what is the reason to use a flow-matching style prediction scheme. Also, the flow-matching's result would be a continuous result, what would be done to convert it to a discrete result as shown in Fig 3?
> >
> > In addition, as the author mentioned, each stage adopt the flow-matching style target. Then for a 8 stage prediction, it would take 8*X total steps for the structure prediction, where X is the number of flow-matching steps for each stage. This could raise huge runtime cost overhead as shown in the newly added Table 4. Therefore, it is questionable for adopting such a time-consuming solution.
> >
> > The reviewer does not think the answer in "Difference as quantization target" address my concern. And the causality does not seem to be reasonable (using more tokens to model error, hence difference is the target). The reviewer wonders if the author can elaborate this part and including a training algorithm of VGS (like what is presented in Sec A, and make it clear the relation between quantization target R, latent Z and content c) in the algorithm part.

---

> > > ### Author Response · Authors · 2025-11-28
> > > **Response to follow up questions**
> > >
> > > Thanks for your clarification and for giving us the chance to discuss further.
> > >
> > > ### 1. The motivation of choosing flow-matching for structure generation
> > >
> > > Thank you for explaining your question.
> > >
> > > **(1) How to obtain discrete results from continuous predictions of flow-matching?**
> > >
> > > Thank you for asking this. We described the operation in Lines 252–256 of the paper.
> > >
> > > At each stage, for every spatial position, we know its structure class from the previous stage. The goal is to split the positions belonging to that class into **two subclasses of equal size**.
> > >
> > > The network output is treated as the **probability** of belonging to one of the two subclasses. We then rank all positions belonging to the same parent class by this probability and select the top half as one subclass and the remaining half as the other.
> > >
> > > In practice, we implement this using Gumbel-top-k, which samples the top half from the joint distribution, rather than using a deterministic ranking. Conceptually, this is similar to classification: although the network outputs continuous probabilities, the final result is a discrete class assignment.
> > >
> > >  (2) **Why using flow-matching even when it seems to be time-consuming**?
> > >
> > > Thanks for this question.
> > >
> > > (a) **Flow-matching is a natural fit for our approach**: First, we believe structure is best modeled globally, not generated locally. Furthermore, there are two specific designs in our structure embedding.
> > >
> > > 1) For visited stages, the structure embedding values are 0 and 2, while unvisited stages are padded with a value of 1.
> > >
> > > 2) During generation, the visited stages are input to the structure generator, meaning the network performs an “inpainting” task, generating only the unvisited stages.
> > >
> > > In this approach:
> > >
> > > - The flow-matching model naturally understands that 1 represents an ungenerated stage, falling between 0 and 2.
> > > - The transition from 1 to 0 or 2 can be interpreted as a flow moving in different directions.
> > > - Random noise naturally introduces diversity into the flow, giving the model different starting points.
> > >
> > > This makes the flow-matching model highly suited to our task.
> > >
> > > (b) **Flow-matching model is more efficient than other methods**: In our initial experiments, we tested different variants for structure generation, but they were less efficient and performed worse than flow-matching.
> > >
> > > Our design goal was to use **a single network** to generate structures at **all stages**, even though inputs and conditions differ across stages.
> > > Because this is a new problem with no existing generation framework designed for our setting, we explored several popular frameworks:
> > > * **(a) One-step generation (similar to the content generator):** simple but produced weak structures.
> > > * **(b) Multi-step masked generation (MaskGIT-style):** this worked well when using *separate* networks for each stage, but we could not make a *single* network handles all stages effectively.
> > > * **(c) Flow matching:** this allowed us to generate high-quality structures for all stages using one unified network.
> > >
> > > Compared to other methods, flow-matching is more efficient when using a single model for all stages.
> > >
> > > We also improved the efficiency of flow-matching model by reducing the model size. The final structure generator is only 1/4 the size of the content generator.
> > > Since the main goal of our work is to propose and generate structural visual granularity, not to optimize speed. Furthermore, our method is already much faster than standard auto-regressive and diffusion models. We did not explore more advanced few-step flow-matching variants, though they can be incorporated easily.

---

> ### Author Response · Authors · 2025-11-28
> **Response to follow up questions (cont.)**
>
> ### 2. **Further clarification on “Difference as quantization target"**
>
> Thanks for your question.
>
> (a) **What is ``quantization target’’?**
>
> This confusion is likely due to our earlier description. In that paragraph, our goal was to explain how we simulate the construction of the visual granularity sequence during generation. We also had a typo: we wrote $x-f_c(x_i)$, but it should be $f_c(x_i) - x_i$, which is the difference between the model output and its input.
>
> When we say “This difference is the quantization target of the current stage,” we mean that $f_c(x_i) - x_i$ approximates the residual  $\mathbf{R}$ in Eq. (5). For this reason, we refer to it as the “quantization target’’ at that stage. No quantization operator is applied at this step.
>
> (b) **Why use the difference as the target?**
>
> As we discussed above, the $f_c(x_i) - x_i$ approximates the residual $\mathbf{R}$ in Eq. (5).
>
> During construction, we use this residual together with the structure map to compute the embedding of each unique token, which is then quantized by selecting a token from the vocabulary.
>
> During generation, after computing this difference, we use the generated structure map to obtain the average feature for each unique token. A linear layer then predicts logits for choosing a token from the vocabulary.
>
> Therefore, in the original paper, we described this difference as the target. We have now revised the paper to make this explanation clearer.

---

### Official Review · Reviewer_enua · 2025-11-01

**Soundness:** 3
**Presentation:** 3
**Contribution:** 2
**Rating:** 4
**Confidence:** 3

**Summary:**

This work introduces content and structure pairs to assist the generation task. In particular, the structure guidance is from the features, without another pretrained network. The structure guidance can improve the generation performance.

**Strengths:**

This work explores the structural information for autoregressive-based image generation. It generates a structural mask by a cluster-based algorithm, without any pretrained method. The experimental results verify its effectiveness.

**Weaknesses:**

- The structured conditional guidance appears to have limited novelty, as similar ideas have been widely explored in text-to-image generation, such as in ControlNet [A].
- There is no ablation study evaluating the effectiveness of the structure and content guidance. In addition, qualitative comparisons with state-of-the-art methods are missing, as are qualitative results for the ablation studies.

[A] Zhang, Lvmin, Anyi Rao, and Maneesh Agrawala. "Adding conditional control to text-to-image diffusion models." ICCV 2023.

**Questions:**

Please see the weakness.

---

> ### Author Response · Authors · 2025-11-20
> **Response to Reviewer enua**
>
> We sincerely appreciate your time and effort in reviewing our paper.
>
> **W1: Limited novelty of structured conditional guidance?**
>
> We respectfully disagree with the comment that our framework has limited novelty or is conceptually similar to ControlNet.
> Below, we highlight the key differences and contributions of our method compared with adding structured conditional guidance, as also discussed in L80–82, L83–88, and L107–123, L126–128:
>
> 1. **Foundational architecture vs. plug-in adapter.** ControlNet [A] is a representative method for adding structured guidance, but it functions as an auxiliary adapter for existing diffusion models. It freezes the base model and attaches a trainable branch to inject spatial control at a fixed granularity.
> In contrast, NVG is a new generation framework. NVG defines a hierarchy of structure granularities and learns structure-aware visual tokens and the full structure and image generation process from scratch. Unlike prior models that rely on uniform patchification (auto-regressive and VAR models) or noise levels (diffusion models), NVG intrinsically ties visual tokens to structure across multiple granularity levels.
> 2. **Intrinsic capability vs. explicit design.**
> ControlNet is explicitly built for conditional generation. In NVG, structure control emerges naturally from the framework rather than being added as a separate component. Moreover, NVG supports multi-granularity structural control, whereas ControlNet only supports a fixed level of control and requires additional pre-trained models.
>
> In summary, NVG is novel and fundamentally different from ControlNet and related control-based methods.
>
> **W2: (a) No ablation study of evaluating the effectiveness of the structure and content guidance?**
>
> This statement is not accurate. First, we want to emphasize that structure and content generation are built-in parts of our framework, not external guidance modules. In addition, we already analyze their impact on the generation results, with qualitative examples shown in Figure 6 and Figure 8 of the original paper. These analyses clearly demonstrate the effectiveness of both the structure and content generation components.
>
> **(b) No qualitative comparison with SOTA?**
>
> Our main goal is to show that images can be generated by explicitly modeling hierarchical visual structures. For this reason, we use qualitative visualizations in Figure 6 and Figure 8 to analyze our generation pipeline. The quantitative comparison with SOTA models is included to show that our framework also achieves strong performance.
> While these results already validate the generation ability and structural modeling effectiveness of our framework, we are happy to provide more qualitative comparisons with other methods. We have added these visualizations in Figure 9 and Figure 10 in the revised version.

---

### Meta-Review · Area_Chair_P3Ng · 2025-12-07

**Summary:**

The paper proposes Next Visual Granularity (NVG), a structured coarse-to-fine image generation framework that separates structure and content across increasing visual granularities. Reviewers broadly acknowledge the novelty, interpretability, and controllability of the approach, and experiments on ImageNet show consistent improvements over strong baselines. The rebuttal-stage revisions to Fig. 4, 9, and 10 make NVG’s advantages over other SOTA methods more evident and improve the overall quality of the manuscript.

The initial scores were **4666**. The authors used the rebuttal to sharpen the narrative and clarify key technical details, and Reviewer bnRk’s follow-up exchange further helped resolve confusion about design choices and training targets, while the other reviewers did not provide rebuttal-stage updates. With the revised figures and added qualitative evidence, the method’s contributions and empirical benefits are communicated more clearly and convincingly.

Therefore, I recommend acceptance.

**Reviewer Concerns:**

The rebuttal addressed Reviewer bnRk’s concerns on method clarity, figures, and the definition of residual and targets through detailed clarifications. Reviewer aGuw’s questions on semantic misalignment, multi-object behavior, and error propagation, and Reviewer 82ro’s requests on robustness and system-level time and memory reporting, were also largely addressed.

**Reviewer Scores:**

My expectation is that all four reviewers would be inclined to maintain their original ratings after the rebuttal. As such, the final scores would likely remain **4666**.

---

### Decision · Program_Chairs · 2026-01-26

Accept (Poster)